# Uncertainty-aware Graph-based Hyperspectral Image Classification

**Linlin Yu[1], Yifei Lou[2], Feng Chen[1]**
[1]Department of Computer Science, University of Texas at Dallas
{linlin.yu,feng.chen}@utdallas.edu
[2]Department of Mathematics and School of Data Sciences and Society,
University of North Carolina at Chapel Hill
yflou@unc.edu

## Abstract

Hyperspectral imaging (HSI) technology captures spectral information across a broad wavelength range, providing richer pixel features compared to traditional color images with only three channels. Although pixel classification in HSI has been extensively studied, especially using graph convolution neural networks (GCNs), quantifying epistemic and aleatoric uncertainties associated with the HSI classification (HSIC) results remains an unexplored area. These two uncertainties are effective for out-of-distribution (OOD) and misclassification detection, respectively. In this paper, we adapt two advanced uncertainty quantification models, evidential GCNs (EGCN) and graph posterior networks (GPN), designed for node classifications in graphs, into the realm of HSIC. We first reveal theoretically that a popular uncertainty cross-entropy (UCE) loss function is insufficient to produce good epistemic uncertainty when learning EGCNs. To mitigate the limitations, we propose two regularization terms. One leverages the inherent property of HSI data where each feature vector is a linear combination of the spectra signatures of the confounding materials, while the other is the total variation (TV) regularization to enforce the spatial smoothness of the evidence with edge-preserving. We demonstrate the effectiveness of the proposed regularization terms on both EGCN and GPN on three real-world HSIC datasets for OOD and misclassification detection tasks. The code is available at GitHub[1].

## 1 Introduction

Hyperspectral (HS) imaging is widely used in various real-world applications including atmospheric science (Saleem et al., 2020), food processing (Ayaz et al., 2020), and forestry (Khan et al., 2020), benefiting from rich spectral information measured at individual pixels. Unlike human eyes which possess only three color receptors sensitive to blue, green, and red channels, HS data provides a wide spectrum of light (visible and near-infrared range) for every pixel in the scene, which enables more faithful classification results compared to traditional classification using color images. As a result, *hyperspectral image classification* (HSIC) attracts considerable research interests (Chen et al., 2014; Ahmad et al., 2017; Hong et al., 2018; Ahmad et al., 2021). Specifically, graph convolution neural network (GCN) (Kipf & Welling, 2017) has found extensive use in HSIC (Shahraki & Prasad, 2018; Qin et al., 2018; Wan et al., 2020; Hong et al., 2020) due to its ability to effectively model the interdependency among pixels (especially when they are far away).

However, there is limited work related to predictive uncertainty quantification for HSIC. For example, it is not practical to assume that all categories (materials) in the scene are known and have available samples for model training. In such scenarios, the model is expected to have the capability to *know what they do not know*, which can be measured by *epistemic uncertainty* from a probabilistic view (uncertainty of model parameters due to limited training data). On the other hand, pixels may be misclassified due to various factors, such as environmental noise, material similarity, and atmospheric effects. Thus, it is desirable for a training model to identify the *unknown what they do not know*, which can be measured by *aleatoric uncertainty* (uncertainty due to randomness). Overall, it is necessary to quantify these two uncertainties to ensure the reliability of HSIC models.

---

[1]https://github.com/linlin-yu/uncertainty-aware-HSIC.git

The epistemic and aleatoric uncertainties can be used to detect out-of-distribution (OOD) pixels that belong to unknown materials and detect pixels that are misclassified to the wrong categories, respectively. OOD detection in HSIC performs in-distribution (ID) classification and OOD detection simultaneously, which is different from *anomaly detection*, as the latter only involves detecting pixels whose spectral characteristics deviate significantly from surrounding or background pixels. Literature has found that epistemic uncertainty is the most effective for OOD detection, while aleatoric uncertainty is most effective for misclassification detection (Zhao et al., 2020; Stadler et al., 2021).

Graph-based models for HSIC construct a graph by regarding each pixel as a node and the interdependency among nodes is defined by an adjacency matrix. As a result, the nodes on the graph are dependent on each other. In contrast to extensive literature for independent inputs (Lakshminarayanan et al., 2017; Gal & Ghahramani, 2016; Charpentier et al., 2022), uncertainty estimation for semi-supervised node classification on a graph with dependent inputs is more complex and thus less explored (Abdar et al., 2021). Notably, two primary investigations have been conducted employing deterministic methodologies. One is the evidential graph convolutional network (EGCN) (Zhao et al., 2020) that extended the evidential neural network (ENN) (Sensoy et al., 2018) on images (independent inputs) to graph data (dependent inputs) with GCNs and graph-based kernel Dirichlet estimation. Throughout the paper, we refer to this model as GKDE or EGCN for brevity. The other is graph posterior network (GPN) (Stadler et al., 2021) that adapted the posterior network (PN) (Charpentier et al., 2020) together with an evidence propagation through the graph nodes. Both GKDE and GPN predict the conjugate prior distribution of categorical distribution, i.e. Dirichlet distribution at each node, and incorporate the uncertainty-cross-entropy (UCE) loss (Biloš et al., 2019) in the overall optimization problem to train the model parameters.

However, the UCE loss function has limitations in effectively learning uncertainty quantification models. First, the UCE-based learning models tend to peak the Dirichlet distribution, thus becoming overly concentrated on the predictive classes in the simplex (feasible) space composed of the class-probability vectors (Bengs et al., 2022). This drawback can be alleviated by introducing entropy-based regularization to encourage the predicted Dirichlet distributions to be uniform (Charpentier et al., 2020; Stadler et al., 2021). Second, it is known empirically that learning models based on UCE alone do not produce accurate epistemic uncertainty for OOD detection, which can be aided by additional regularization terms, e.g., the aforementioned GKDE (Zhao et al., 2020). However, we show in our experiments both GKDE and GPN do not have satisfactory results in HSIC.

In this work, we consider the uncertainty quantification task for graph-based hyperspectral image classification. **Our Contributions** are summarized as follows. **First**, we characterize scenarios when minimizing the UCE loss fails to provide accurate epistemic uncertainty estimation. In particular, minimizing the UCE loss does not help an EGCN to learn embeddings that are capable of mapping OOD nodes into the detectable region near the decision boundary. **Second**, we propose a multidimensional uncertainty estimation framework for HSIC. To the best of our knowledge, this is a pioneer work in discussing the uncertainty estimation on the graph-based HSIC models. **Third**, we introduce a physics-guided unmixing-based regularization (UR) to address the shortcomings of the UCE loss when quantifying epistemic uncertainty. Here, we assume that OOD pixels are mostly composed of an unknown material and the UR term is the reconstruction squared loss for decomposing into the ID materials and the OOD material. **Fourth**, we adopt the TV term to propagate predicted evidence along the decision boundary (not across), thus preserving spatial edges between ID and OOD nodes. **Finally**, we present extensive empirical experiments to demonstrate the effectiveness of the proposed regularization terms on both EGCN and GPN, using three real-world HSIC datasets for OOD and misclassification detection, in comparison to five baselines.

## 2 PRELIMINARY

### 2.1 GRAPH-BASED HYPERSPECTRAL IMAGE CLASSIFICATION (HSIC)

HSIC aims to assign a unique label to each pixel based on its spectral and spatial properties. The input HS data can be represented as $\boldsymbol{X} = [\boldsymbol{x}^1, \boldsymbol{x}^2, \cdots, \boldsymbol{x}^{(HW)}] \in \mathbb{R}^{(H \times W) \times B}$, where $B$ is the number of spectral bands (feature dimension) and $H \times W$ is the spatial dimension. Letting $N = HW$, we stack the 2D spatial domain to a vector, and hence each pixel $i$ is associated with a feature vector $\boldsymbol{x}^i \in \mathbb{R}^B$, $\forall i \in [N]$. For classification purposes, each pixel $i$ has a class label $y^i \in [C]$ associated with a specific constituent material, where $C$ is the number of classes known as *a priori*.

The graph-based HSIC technique (Qin et al., 2020) builds a graph, in which each vertex corresponds to a pixel in the 2D spatial domain and the weighted adjacency matrix $\mathbb{A}$ is calculated based on similarities between node-level features: i.e.,

$$\mathbb{A}_{ij} = \exp(-d(\mathbf{x}^i, \mathbf{x}^j)/\sigma), \quad \forall i, j \in [N], \tag{1}$$

where $d(\mathbf{x}^i, \mathbf{x}^j)$ is the Euclidean distance (or cosine similarity) between vertices $i$ and $j$, and $\sigma$ is tuned to optimize how similar two nodes are. Suppose the graph is defined as $\mathcal{G}(\mathbb{V}, \mathbb{E}, \boldsymbol{X}, \boldsymbol{Y}_\mathbb{L})$, where $\mathbb{V} = \{1, \cdots, N\}$ is a ground set of nodes, $\mathbb{E} \subseteq \mathbb{V} \times \mathbb{V}$ is a ground set of edges, $\boldsymbol{X} = [\boldsymbol{x}^1, \boldsymbol{x}^2, \cdots, \boldsymbol{x}^N] \in \mathbb{R}^{N \times B}$ is the node-level feature matrix, $\boldsymbol{x}^i \in \mathbb{R}^B$ is the feature vector of node $i$, $\boldsymbol{Y}_\mathbb{L} = \{y^i | i \in \mathbb{L}\} \in \mathbb{R}^{|\mathbb{L}|}$ is the label for the training node set $\mathbb{L} \subset \mathbb{V}$, and $y^i \in [C]$ is the label for node $i$. The GCN-based HSIC method (Hong et al., 2020) is formulated as $[\mathbf{p}^i]_{i \in \mathbb{V}} = f(\mathbb{A}, \mathbf{X}; \boldsymbol{\theta})$, where $\mathbf{p}^i$ is the probability vector of node $i$ and $f(\cdot)$ is a standard GCN function that depends on the adjacency matrix $\mathbb{A}$, the data matrix $\mathbf{X}$, and a set of network parameters, denoted by $\boldsymbol{\theta}$.

## 2.2 EVIDENTIAL GRAPH CONVOLUTIONAL NETWORKS FOR NODE CLASSIFICATION

An evidential GCN (EGCN) (Zhao et al., 2020) takes graph $\mathcal{G}$ as INPUT and predicts an evidence vector $\mathbf{e}^i = [e_1^i, \cdots, e_C^i]$ for each node $i$ as OUTPUT: $[\mathbf{e}^i]_{i \in \mathbb{V}} = f(\mathbb{A}, \mathbf{X}; \boldsymbol{\theta})$, where $e_c^i$ is a measure of the amount of support collected form the training labels $\mathbf{Y}_\mathbb{L}$ in favor of node $i$ to be classified to the class $c$. EGCN is the same as a classical GCN, except that the activation function (e.g., exponential or ReLU) of the output layer is unbounded, outputting an evidence vector, instead of the softmax function outputting class probabilities. The evidence vector can quantify predictive uncertainty through a well-defined theoretical framework called subjective logic (SL) (Jsang, 2018). More specifically, a multinomial opinion $\omega = (\mathbf{b}, u)$ in SL can be defined as:

$$b_c = \frac{e_c}{S} \text{ and } u = \frac{C}{S}, \text{ for } c = 1, \cdots, C, \tag{2}$$

where $\mathbf{b} = [b_1, \cdots, b_C]^T$ represents the beliefs of the $C$ classes, $u$ is the uncertainty mass representing *the vacuity of evidence*, and $S = \sum_{c=1}^C (e_c + 1)$. It is straightforward that $b_c \geq 0, u \geq 0$, and $\sum_{c=1}^C b_c + u = 1$. A multinomial opinion $\omega$ can be equivalently represented by a Dirichlet distribution: $\mathbf{p} \sim \text{Dir}(\boldsymbol{\alpha})$, where $\mathbf{p} = [p_1, \cdots, p_C]$ is a probability vector of C classes and $\boldsymbol{\alpha} = [\alpha_1, \cdots, \alpha_C]$ are called *concentration parameters* with $\alpha_c = e_c + 1$. The class label $\mathbf{y}^i$, probability vector $\mathbf{p}^i$, and the evidence vector $\mathbf{e}^i$ for node $i$ have the following probabilistic relations:

$$\mathbf{y}^i \sim \text{Cat}(\mathbf{p}^i), \ \ \mathbf{p}^i \sim \text{Dir}(\mathbf{p}^i | \boldsymbol{\alpha}^i), \ \ \boldsymbol{\alpha}^i = \mathbf{e}^i + \mathbf{1}, \ \ [\mathbf{e}^i]_{i \in \mathbb{V}} = f(\mathbb{A}, \mathbf{X}; \boldsymbol{\theta}). \tag{3}$$

The expected class probability is equal to the mean of the Dirichlet distribution, i.e. $\bar{\boldsymbol{p}} = \frac{\boldsymbol{\alpha}}{S}$ in the sense that $S$ can also be defined by $S = \sum_{c=1}^C \alpha_c$. Based on evidential theory, a lack of evidence, e.g., "I don't know," can be expressed by a close-to-one vacuity $u$ (or a uniform Dirichlet).

An EGCN is trained based on the uncertainty cross-entropy (UCE) loss function, defined by

$$\text{UCE}(\boldsymbol{\alpha}^i, \mathbf{y}^i; \boldsymbol{\theta}) = \mathbb{E}_{\boldsymbol{p}^i \sim \text{Dir}(\boldsymbol{p}^i | \boldsymbol{\alpha}^i)} \left[ -\log \mathbb{P}(\boldsymbol{y}^i | \boldsymbol{p}^i) \right], \tag{4}$$

which can be interpreted as the expectation of the standard cross-entropy loss with respect to the distribution of class probabilities: $\boldsymbol{p}^i \sim \text{Dir}(\boldsymbol{p}^i | \boldsymbol{\alpha}^i)$. Alternatively, Stadler et al. (2021) proposed a new network architecture (as opposed to a classical GCN architecture), namely graph posterior networks (GPN), to predict node-level Dirichlet distributions. Specifically, GPN consists of three modules: multilayer perceptron (MLP) layers for node-level feature embedding, a normalizing flow module to estimate node-level densities in the embedded space, and a personalized page rank propagation layer (Gasteiger et al., 2018) to smooth the concentration parameters among neighboring nodes.

## 2.3 UNCERTAINTY QUANTIFICATION

Aleatoric uncertainty is the uncertainty in the class prediction, which is measured by the entropy of categorical distribution(Malinin et al., 2017), i.e. $u^{\text{alea}} = \mathbb{H}(\text{Cat}(\bar{\boldsymbol{p}}))$ or *confidence* (Charpentier et al., 2020), i.e., $u^{\text{alea}} = -\max_c \bar{p}_c$. It exhibits higher values when the categorical distribution is flat. In contrast, epistemic uncertainty is the uncertainty on categorical distribution and can be measured by the total evidence count, i.e. $u^{\text{epis}} = C/S$, which is referred to as *vacuity* from the viewpoint of evidential uncertainty (Josang et al., 2018). When the distribution of categorical distribution, which is the Dirichlet distribution in the evidential-based models, is spread out, the epistemic uncertainty is high. Aleatoric uncertainty is proven to be effective for detecting misclassifications while epistemic is often used to identify OOD samples (Zhao et al., 2020).

## 3 UNCERTAINTY-AWARE REGULARIZED LEARNING

### 3.1 LIMITATIONS OF UCE AND EXISTING REGULARIZATION TECHNIQUES

Without the loss of generality, we focus on binary classification tasks throughout this section for theoretical analysis; the generalization to multiple classes can be analyzed similarly to Collins et al. (2023) and Kristiadi et al. (2020). A typical EGCN architecture has several graph convolutional (GC) layers followed by one MLP layer (Zhao et al., 2020). GC layers produce node-level embeddings to capture the graph dependency among the nodes in the sense that nodes that are neighbors in the graph are more likely to be spatial neighbors in the embedded space, denoted by $\mathcal{D} \subset \mathbb{R}^D$. Specifically for homophily graphs (Ma et al., 2021), GC layers can generate embeddings that can separate different classes. Note that the three HSIC datasets used in our experiments are indeed homophily graphs, as discussed in Appendix B. The MLP layer in EGCN helps to reduce the dimensions of the embedded space while producing node-level evidence. We demonstrate that the MLP layer learned based on UCE fails to produce accurate evidence predictions, even in the ideal case where the GC layers can produce perfectly separable node embeddings. Let $\mathbf{z}^i \in \mathbb{R}^D$ denote the embedded vector of node $i \in \mathbb{V}$. The MLP layer for node-level evidence prediction can be formulated as:

$$\mathbf{e}(\mathbf{z}; \boldsymbol{\theta}) := [e_+(\mathbf{z}; \boldsymbol{\theta}), e_-(\mathbf{z}; \boldsymbol{\theta})] = [\sigma(\mathbf{w}^T \mathbf{z} + b), \sigma(-\mathbf{w}^T \mathbf{z} - b)], \tag{5}$$

where $\boldsymbol{\theta} = \{\mathbf{w}, b\}$, $\mathbf{w} \in \mathbb{R}^D$, $b \in \mathbb{R}$, and $\sigma(\cdot)$ is the activation function (e.g. ReLU and exponential) that outputs evidence values. We start with the lower and upper bounds for MLP-based ENNs.

**Proposition 1.** *Suppose $\mathbf{z} \in \mathcal{D} \subset \mathbb{R}^D$ is a point in the embedded space and $y \in \{-1, +1\}$ is its binary class label. An MLP-based ENN has the lower and upper bounds for the UCE loss:*

$$\frac{1}{e_y(\mathbf{z}; \boldsymbol{\theta}) + 1} \le UCE(\boldsymbol{\alpha}(\mathbf{z}; \boldsymbol{\theta}), y; \boldsymbol{\theta}) \le \frac{\lceil e_{-y}(\mathbf{z}; \boldsymbol{\theta}) \rceil + 1}{e_y(\mathbf{z}; \boldsymbol{\theta})}, \tag{6}$$

*where $e_y(\mathbf{z}; \boldsymbol{\theta})$ is the evidence of classs $y$, $\boldsymbol{\alpha}(\mathbf{z}; \boldsymbol{\theta}) = \mathbf{e}(\mathbf{z}; \boldsymbol{\theta}) + 1$, and $\lceil \cdot \rceil$ denotes the ceiling operator. If ENN can predict $y$ correctly: $y(e_+(\mathbf{z}; \boldsymbol{\theta}) - e_-(\mathbf{z}; \boldsymbol{\theta})) > 0$, we have a tighter upper bound:*

$$UCE(\boldsymbol{\alpha}(\mathbf{z}), y; \boldsymbol{\theta}) \le \overline{UCE}(\boldsymbol{\alpha}(\mathbf{z}), y; \boldsymbol{\theta}) := \frac{r + 1}{e_y(\mathbf{z}; \boldsymbol{\theta})}, \tag{7}$$

*where $r = 0$ for the ReLU activation function and $r = 1$ for the exponential activation function.*

Please refer to Appendix A.1 for the proof of Proposition 1. Note that the upper bound in (7) is tight under the universal approximation theorem (Pinkus, 1999). Specifically, the optimal parameter $\boldsymbol{\theta}^\star$ that minimizes the UCE on a training set has the property: $e_y(\mathbf{z}; \boldsymbol{\theta}^*) \to \infty$ and $e_{-y}(\mathbf{z}; \boldsymbol{\theta}^*) \to 0$, as demonstrated in Lemma 2 in Appendix A.1. Therefore, we have

$$\lim_{e_y(\mathbf{z};\boldsymbol{\theta})\to\infty} |\frac{r+1}{e_y(\mathbf{z};\boldsymbol{\theta})} - \text{UCE}(\boldsymbol{\alpha}, y; \boldsymbol{\theta})| \le \lim_{e_y(\mathbf{z};\boldsymbol{\theta})\to\infty} \frac{r+1}{e_y(\mathbf{z};\boldsymbol{\theta})} - \frac{1}{e_y(\mathbf{z};\boldsymbol{\theta})+1}$$

$$\le \lim_{e_y(\mathbf{z};\boldsymbol{\theta})\to\infty} \frac{r \cdot e_y(\mathbf{z};\boldsymbol{\theta}) + r + 1}{(e_y(\mathbf{z};\boldsymbol{\theta})+1)e_y(\mathbf{z};\boldsymbol{\theta})} = 0.$$

Next, we establish in Theorem 1 that the optimal solution when minimizing the upper bound $\overline{\text{UCE}}(\boldsymbol{\alpha}(\mathbf{z}), \mathbf{y}; \boldsymbol{\theta})$ defined in Equation (7) with exponential activation function $\sigma(\cdot)$ has a closed-form expression that is equivalent to the optimal solution of linear discriminative analysis (LDA) under certain assumptions.

**Theorem 1.** *We assume that (i) feature vectors belonging to classes $\{\pm 1\}$ follow Gaussian distributions with the same covariance matrix and the means $\pm\boldsymbol{\mu}$, respectively, i.e., $\mathbb{P}(\mathbf{z}, y) = \mathbb{P}(y = +1)\mathcal{N}(\mathbf{z}; \boldsymbol{\mu}, \boldsymbol{\Sigma}) + \mathbb{P}(y = -1)\mathcal{N}(\mathbf{z}; -\boldsymbol{\mu}, \boldsymbol{\Sigma})$, with $\mathbb{P}(y = +1) = \mathbb{P}(y = -1) = 0.5$; (ii) the optimal solutions $\boldsymbol{\theta}^\star$ that minimize $\mathbb{E}_{(\mathbf{z},y)\sim\mathbb{P}(\mathbf{z},y)}\overline{UCE}(\boldsymbol{\alpha}(\mathbf{z}), \mathbf{y}; \boldsymbol{\theta})$ can linearly separate both classes: $e_y(\mathbf{z}; \boldsymbol{\theta}) > e_{-y}(\mathbf{z}; \boldsymbol{\theta}), \forall (\mathbf{z}, y)$. Let $\sigma(\cdot)$ be the exponential function. The optimal solution $\boldsymbol{\theta}^\star = (\mathbf{w}^\star, b^\star)$ is the same as the optimal solution of LDA, i.e.,*

$$\mathbf{w}^\star = \boldsymbol{\Sigma}^{-1}\boldsymbol{\mu} \quad and \quad b^\star = 0. \tag{8}$$

Theorem 1 has several important implications when the classes are separable. **First**, the MLP layer in EGCN learned based on $\overline{\text{UCE}}$ has the same objective as in LDA: Finding a projection that maximizes the separation between the projected class means with a small variance within each class. Unfortunately, as illustrated in Fig. 1, this objective does not help learn a projection that maps OOD data points to the **grey region of low evidence** near the decision boundary, where GCNs can effectively detect OODs: $e_y(\mathbf{z}; \boldsymbol{\theta}^\star) = 0$ (or equivalently $\mathbf{w}^{\star T}\mathbf{z} + b^\star = 0$).

For any far-away OOD data point $\tilde{\mathbf{z}} = \delta \cdot \mathbf{z}$, where $\delta \to \infty$ and $(\mathbf{z}, y) \in \mathbb{P}(\mathbf{z}, y)$ is an ID data point, the predicted evidence approaches $+\infty$ (or equivalently the epistemic uncertainty approaches 0): $e_y(\tilde{\mathbf{z}}; \boldsymbol{\theta}^\star) = \exp(y(\mathbf{w}^{\star T}\tilde{\mathbf{z}} + b^\star)) = \exp(y\mathbf{w}^{\star T}\delta\mathbf{z}) = (\exp(y\mathbf{w}^{\star T}\mathbf{z}))^\delta \to \infty$, when $\mathbf{z}^T\boldsymbol{\Sigma}^{-1}\boldsymbol{\mu} \neq 0$, given that $b^\star = 0$ and $\exp(y(\mathbf{w}^{\star T}\tilde{\mathbf{z}} + b^\star)) = \exp(y\mathbf{w}^{\star T}\tilde{\mathbf{z}}) > 1$. **Second**, the evidence predictions for the testing data points are not influenced by the distance between the two class means, as it is not a factor in $\mathbf{w}^\star$ and $b^\star$. **Third**, Fig. 1 shows that we can identify the light-blue and light-grey OOD regions of different characteristics in the feature space of $\mathbf{z}$ based on the projection $\mathbf{w}^\star = \boldsymbol{\Sigma}^{-1}\boldsymbol{\mu}$. In particular, the learned MLP layer predicts higher evidence for OOD nodes in the light-blue region than the one of ID nodes; and predicts evidence similar to those of ID data points for OOD data points in the light-grey region. The learned MLP can only predict small evidence for OOD points in the small light-grey region near the decision boundary: $\mathbf{w}^{\star T}\mathbf{z} + b^\star = 0$.

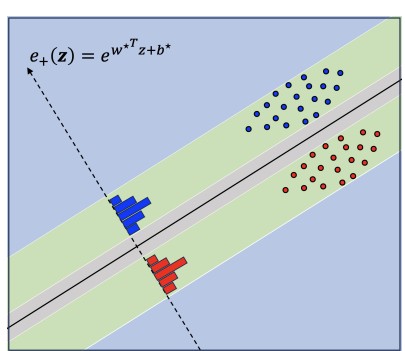

Figure 1: Two-separable-class case: The grey region in the feature space is a detectable OOD region by one-layer MLP-based ENN learned using UCE. The light-blue and light-green OOD regions are not detectable by ENN.

We remark on several assumptions in Theorem 1. First, the assumption on the Gaussian means $\boldsymbol{\mu}$ and $-\boldsymbol{\mu}$ can always be true by translating the origin of the feature space to the middle point of two centers as a preprocessing step. Second, we assumed the same covariance matrix $\boldsymbol{\Sigma}$ for the two Gaussians to obtain an analytical solution in Eq. (8). For different covariance matrices, the optimal solution is non-identical to that of LDA. Third, we assume that the classes are linearly separable so that the MLP layer can be defined in Eq. (5). The linear separability has been assumed in OOD-related theoretical analysis such as Ahuja et al. (2021). For the non-separable case, the MLP layer is defined as $\mathbf{e}(\mathbf{z}; \boldsymbol{\theta}) := [e_+(\mathbf{z}; \boldsymbol{\theta}), e_-(\mathbf{z}; \boldsymbol{\theta})] = [\sigma(\mathbf{w}_1^T\mathbf{z} + b_1), \sigma(\mathbf{w}_2^T\mathbf{z} + b_2)]$, where the weight and bias parameters for predicting the evidence values of the two classes are different. As demonstrated in Fig. 2, the grey, light-green, and light-blue regions have more complex

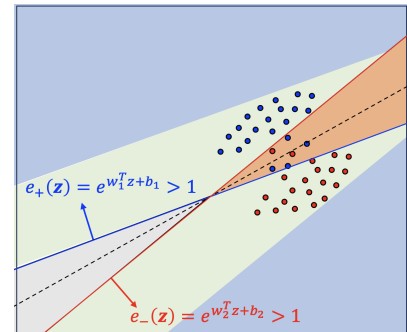

Figure 2: Two-non-separable-class case. The grey, light-green, and light-blue regions are the same as those in Fig.1. In the orange region, the predicted evidence values of both classes are larger than 1.

shapes compared to the separable case in Fig. 1. The OOD points that can be detected by the MLP layer are within the grey region: $\{\mathbf{z}|e_y(\mathbf{z}; \boldsymbol{\theta}) < 1, e_{-y}(\mathbf{z}; \boldsymbol{\theta}) < 1\}$. Further, there is an orange region for the non-separable case: $\{\mathbf{z}|e_y(\mathbf{z}; \boldsymbol{\theta}) > 1, e_{-y}(\mathbf{z}; \boldsymbol{\theta}) > 1\}$, in which the evidence values for both classes are larger than 1. Our theoretical results on EGCN may not be generalizable to GPN. GPN predicts evidence values based on density estimation in the embedded space instead of MLP layers as used in EGCN.

We demonstrate that minimizing the UCE loss does not help to learn the MLP layer to map the OOD regions (e.g., light-green and light-green OOD regions in Figs. 1 and 2) into the detectable OOD region of EGCN near the decision boundary. Zhao et al. (2020) proposed to use the KL divergence-based regularization term: $\sum_{i \in \mathbb{L}} \text{KL}(\text{Dir}(\hat{\boldsymbol{p}}^i|\hat{\boldsymbol{\alpha}}^i), \text{Dir}(\boldsymbol{p}^i|\boldsymbol{\alpha}^i))$, to enforce the closeness between $\hat{\boldsymbol{\alpha}}^i$ and $\boldsymbol{\alpha}^i$, $\forall i \in \mathbb{V}$, where $\hat{\boldsymbol{\alpha}}^i$ is a pre-computed teacher based on graph-kernel distance. However, this term assumes that OOD test nodes are far away in terms of graph-based distance from the training (ID) nodes compared to ID test nodes. This assumption is not always valid based on our empirical performance of the GKDE teacher on three HSIC datasets in Appendix E.5.

## 3.2 Unmixing and Evidence-based Uncertainty Regularizations

Due to the limited spatial resolution of HSI sensors, it is conceivable that each pixel in HSI data may contain a combination of materials, and hence it is desirable to decompose a single pixel into the proportions of constituent materials (a.k.a. abundance) (Iordache et al., 2012). We assume that there exist $C$ pure materials (a.k.a. endmembers) in the scene, each with the corresponding signature $\boldsymbol{m}_c \in \mathbb{R}^B, c \in \{1, \cdots, C\}$. A matrix formed by all of these signatures is called a mixing matrix, denoted by $\boldsymbol{M} = [\boldsymbol{m}_1, \ldots, \boldsymbol{m}_C] \in \mathbb{R}^{B \times C}$. The abundance map obtained by the abundance coefficients of all the pixels can be represented by a matrix $\boldsymbol{V} \in \mathbb{R}^{C \times N}$. We adopt a linear mixing model in which the spectral measurement at each pixel is a linear combination of the endmembers, i.e.,

$$\boldsymbol{x}^i = \sum\nolimits_{c=1}^{C} v_c^i \boldsymbol{m}_c + \boldsymbol{\eta}^i, \tag{9}$$

where $v_c^i$ is the abundance coefficient for the $c$-th material at the $i$th pixel and $\boldsymbol{\eta}^i$ denotes a noise term. Denote $\boldsymbol{v}^i = [v_1^i, \ldots, v_C^i]$. It is typical to assume $\sum_{c=1}^{C} v_c^i = 1$, as each abundance vector resides within the probability simplex. We adopt the linear mixing model (9) for its simplicity. There are more complicated nonlinear models by taking into account endmember-wise scaling factors (Drumetz et al., 2016), spectral variability (Hong et al., 2018), and illumination-induced variability (Drumetz et al., 2019). Decomposing the HS data $\mathbf{X}$ into a collection of reference spectral signatures $\mathbf{M}$ with associated abundance matrix $\mathbf{V}$ is referred to as *hyperspectral unmixing*.

**Unmixing-based Regularization (UR).** HSIC is related to hyperspectral unmixing in that the in-distribution (ID) classes are associated with the $C$ known endmembers. Under the OOD detection setting where OOD classes are associated with unknown materials, we can consider a linear mixing model, where the signatures of the ID materials, $\{\mathbf{m}_1, \cdots, \mathbf{m}_C\}$ are given. We assume that OOD nodes are associated with the same unknown material that is denoted by $\mathbf{m}_o$. The hyperspectral unmixing problem can be formulated as

$$\min_{\boldsymbol{m}_o, \boldsymbol{v}^i, v_o^i} \sum\nolimits_{i \in \mathbb{V}} \|\boldsymbol{x}^i - \boldsymbol{M}\boldsymbol{v}^i - v_o^i \boldsymbol{m}_o\|_2^2. \tag{10}$$

We propose to use the beliefs $\mathbf{b}^i(\boldsymbol{\theta})$ and the vacuity $u^i(\boldsymbol{\theta})$ (the epistemic uncertainty measure) to approximate the abundance coefficients $\boldsymbol{v}^i$ of ID materials and the abundance coefficient $v_o^i$ of the OOD material, respectively. The rationale of such approximations, $\mathbf{b}^i(\boldsymbol{\theta}) \approx \boldsymbol{v}^i(\boldsymbol{\theta})$ and $u^i(\boldsymbol{\theta}) \approx v_o^i(\boldsymbol{\theta})$, is threefold. First, the sum-to-one property on beliefs and vacuity: $\sum_{c=1}^{C} b_c^i + u^i = 1$, is aligned with the one on abundance coefficients: $\sum_{c=1}^{C} v_c^i + v_o^i = 1$. Second, the vacuity $u^i$ for ID node $i$ should be close to zero, and hence the beliefs are analogous to class probabilities (Jsang, 2018), which can be used to approximate the abundance coefficients (Chen et al., 2023). Third, the vacuity for an OOD node is close to one and its belief is close to zero, implying that the abundance coefficient $v_o^i(\boldsymbol{\theta})$ should be close to one and $\boldsymbol{v}^i(\boldsymbol{\theta})$ be close to $\mathbf{0}$. Using the approximations, we turn the unmixing problem (10) into an unmixing regularization (UR) term,

$$\min_{\boldsymbol{m}_o, \boldsymbol{\theta}} \mathrm{UR}(\boldsymbol{m}_o, \boldsymbol{\theta}) := \sum\nolimits_{i \in \mathbb{V}} \|\boldsymbol{x}^i - \boldsymbol{M}\boldsymbol{b}^i(\boldsymbol{\theta}) - u_o^i(\boldsymbol{\theta})\boldsymbol{m}_o\|_2^2, \tag{11}$$

where $\boldsymbol{b}^i, u_o^i (\forall i \in \mathbb{V})$ can be derived by the evidence $\boldsymbol{e}^i(\boldsymbol{\theta}) = f_i(\mathbb{A}, \mathbf{X}; \boldsymbol{\theta})$, or $\boldsymbol{e}^i$ for brevity. Minimizing the UR term encourage high vacuity for OOD nodes and low vacuity for ID ones. Given $\boldsymbol{\theta}$, there is a closed-form solution for the optimal $\boldsymbol{m}_o$, i.e.,

$$\boldsymbol{m}_o^* = \frac{\sum_{i \in \mathbb{V}} u_o^i(\boldsymbol{\theta})(\boldsymbol{x}^i - \boldsymbol{M}\boldsymbol{b}^i(\boldsymbol{\theta}))}{\sum_{i \in \mathbb{V}} (u_o^i(\boldsymbol{\theta}))^2}. \tag{12}$$

Please refer to Appendix A.3 for more details. Using the definitions of $\boldsymbol{b} = \frac{\boldsymbol{e}}{C + \sum_{c=1}^{C} e_c}$, $u = \frac{C}{C + \sum_{c=1}^{C} e_c}$, we rewrite $\mathrm{UR}(\boldsymbol{m}_o^*, \boldsymbol{\theta})$ with respect to evidence $\boldsymbol{e}$, i.e.,

$$\mathrm{UR}(\boldsymbol{e}) = \sum\nolimits_{i \in \mathbb{V}} \|\boldsymbol{x}^i - \frac{\sum_{c=1}^{C} e_c^i \boldsymbol{m}_c}{C + \sum_{c=1}^{C} e_c^i} - \frac{C\boldsymbol{m}_o^*}{C + \sum_{c=1}^{C} e_c^i}\|_2^2. \tag{13}$$

**Proposition 2.** *Assume the linear model (9) holds without noise, the gradient descent for minimizing the UR regularization increases the predicted evidence of ground-truth class for ID instances and decrease the total evidence for OOD instances with the corresponding pure material contained in the pixel. Formally, we have*

(a) *For an instance $(\boldsymbol{x}^i, y^i)$ with feature matrix $\boldsymbol{x}^i = \boldsymbol{m}_{y^i}$ and $y^i \in \{1, \ldots, C\}$ is the ground truth label, one has*

$$\frac{\partial UR(\boldsymbol{e})}{\partial e_{y^i}^i} \leq 0. \tag{14}$$

(b) *For an OOD instance instance $(\boldsymbol{x}^i, y^i)$ with $\boldsymbol{x}^i = \boldsymbol{m}_o^*$ and $y^i = o \notin \{1, \ldots, C\}$*

$$\sum_{c=1}^{C} \frac{\partial UR(\boldsymbol{e})}{\partial e_c^i} \geq 0. \tag{15}$$

We present two desired properties of the UR term in Proposition 2. Part (a) is consistent with minimizing the UCE loss for ID nodes, aiming to predict high class-wise evidence for ground truth class. This often results in an increased total evidence for ID nodes during training iterations. Part (b) implies a decrease in total evidence for OOD samples when minimizing the UR term, resulting in a higher vacuity score, which provides additional information beyond the UCE loss. It is the inherent physical characteristics of hyperspectral data that implicitly help distinguish OOD and ID. Specifically, each pixel in a hyperspectral image contains a spectrum, which is a mixture of the spectra of all materials present in that pixel. ID and OOD pixels naturally contain different materials. Note that Proposition 2 is agnostic to model architectures, i.e., the UR term can be applied to any uncertainty quantification architectures.

**Evidence-based Total Variation Regularization (TV).** The total variation (TV) regularization (Iordache et al., 2012) was applied to the abundance coefficients to enforce the spatial smoothness while preserving edges for hyperspectral unmixing. As the graph $\mathcal{G}$ does not incorporate spatial information, we propose the use of TV on the node-level vacuity value, which is inversely proportional to the Dirichlet level strengths (or equivalently total evidence). To define the discrete TV regularization, we represent a 2D image of size $H \times W$ as a vector via a linear indexing, i.e., $((h-1)H + m)$-th component denotes the location at $(h, m)$. Define two matrices $D_x, D_y$ to be the finite forward difference operators with periodic boundary conditions in the horizontal and vertical directions, respectively. Then the discrete form of the (anisotropic) TV norm is defined by

$$\text{TV}(\boldsymbol{u}) = \|D_x \boldsymbol{u}\|_1 + \|D_y \boldsymbol{u}\|_1. \tag{16}$$

**Regularized Learning.** Putting together, we formulate the regularized learning objective as,

$$\mathcal{L}(\boldsymbol{\theta}, \boldsymbol{m}_o) = \sum\nolimits_{i \in \mathbb{L}} \Big( \text{UCE}(\boldsymbol{\alpha}^i, \mathbf{y}^i; \boldsymbol{\theta})) + \lambda_1 R(\boldsymbol{\theta}) \Big) + \lambda_2 \text{UR}(\boldsymbol{\theta}, \boldsymbol{m}_o) + \lambda_3 \text{TV}(\boldsymbol{u}(\boldsymbol{\theta})), \tag{17}$$

where $R(\boldsymbol{\theta})$ refers to the model (GKDE or GPN)-specific regularization term and $\lambda_1, \lambda_2, \lambda_3$ are hyperparameters. For GPN, $R(\boldsymbol{\theta}) = \sum_{i \in \mathbb{L}} \text{ENT}(\text{Dir}(\boldsymbol{p}^i | \boldsymbol{\alpha}^i))$. The GKDE regularization term can be found in the last paragraph of Section 3.1. The TV term is applied on the vacuity score $\boldsymbol{u}(\boldsymbol{\theta})$. The last two terms only require node features and are applied to the whole graph $\mathbb{V}$. The model parameter $\boldsymbol{\theta}$ and $\boldsymbol{m}_o$ in UR term can be optimized alternatively: closed-form solution for $\boldsymbol{m}_o$ in (12) and gradient descent to update the model parameters $\boldsymbol{\theta}$.

## 4 EXPERIMENT

### 4.1 EXPERIMENT SETUP

**Datasets** We use three HSIC datasets for evaluation: the University of Pavia (UP), the University of Houston (UH), and the Kennedy Space Center (KSC). For train/(validation + test) split, we adopt the public challenge split for UH (Debes et al., 2014), the same split for UP as (Hong et al., 2020), and a random split for KSC with 20 nodes for training. For validation/test split, we use 0.2/0.8. The number of disjoint train/validation/test samples selected from each class used for all the experimental results is presented in Appendix B.

**Competing Schemes** We consider two state-of-the-art uncertainty quantification backbones designed for graph data: EGCN (Zhao et al., 2020) and GPN (Stadler et al., 2021). For EGCN, we include GKDE regularization by default. Softmax-GCN (Kipf & Welling, 2017) is a classic GCN for semi-supervised node classification and uses the softmax as the last activation layer. We

use the entropy as the uncertainty score as (Hendrycks & Gimpel, 2016). Though this paper focuses on OOD detection, we include three anomaly detection models: TLRSR (Wang et al., 2022), RGAE (Fan et al., 2021), and TRDFTVAD (Feng et al., 2023) as baselines. We use the features of all the nodes on the graph for anomaly detection and regard the OOD nodes as anomalies. The detailed settings along with parameter tuning are presented in Appendix D. We evaluate the performance via the area under the Receiver Operating Characteristic (AUROC) curve and the area under the Precision-Recall (AUPR) curve with epistemic uncertainty score in OOD detection tasks and aleatoric uncertainty score in misclassification detection tasks. We report the mean and standard deviation over five random trials in each table.

## 4.2 RESULTS

**Misclassification detection.** The problem of misclassification detection is to identify whether a given prediction is misclassified or not with estimated uncertainty scores. A misclassified prediction is given a positive label (i.e.,1), while a correct prediction is given a negative label (i.e., 0). Table 1 shows the misclassification detection results where the best performance over all the models is highlighted in bold. We observe that softmax-GCN is decent on misclassification detection, indicating that misclassified nodes tend to have predicted class probabilities spread out across ID categories and entropy can capture reasonable aleatoric uncertainty for deterministic softmax models. Besides, our proposed uncertainty quantification frameworks show comparable results of the misclassification detection task compared to softmax-GCN on UP and UH, while better on KSC.

Table 1: AUROC and AUPR for the misclassification detection.

| dataset | UP | | UH | | KSC | |
|---|---|---|---|---|---|---|
| | AUROC | AUPR | AUROC | AUPR | AUROC | AUPR |
| softmax-GCN | **78.95**±1.18 | 47.58±0.78 | **89.22**±0.25 | 52.85±1.62 | 89.22±0.25 | 52.85±1.62 |
| EGCN | 77.37±0.61 | 47.78±0.44 | 87.98±0.50 | 76.18±0.82 | 90.18±0.32 | 54.38±1.59 |
| EGCN - UR | 77.89±1.37 | 47.16±0.52 | 88.47±0.56 | **76.77**±1.15 | 90.21±0.38 | 53.51±1.79 |
| EGCN - UR - TV (**Ours**) | 78.83±0.85 | **48.73**±0.49 | 87.96±0.49 | 75.76±1.51 | **90.37**±0.54 | 55.09±1.71 |
| GPN | 73.39±0.70 | 44.38±0.60 | 80.66±0.95 | 67.24±1.92 | 78.13±13.82 | 54.68±7.69 |
| GPN - UR | 73.58±0.36 | 45.02±0.42 | 81.08±1.05 | 67.25±1.24 | 82.38±6.78 | **55.28**±6.17 |
| GPN - UR - TV (**Ours**) | 73.35±0.33 | 47.99±2.37 | 83.36±0.68 | 67.57±1.83 | 78.23±6.58 | 53.08±8.09 |

The bold numbers are the best results over all models. The underlined numbers are the best results within the same model type.

**OOD detection.** OOD detection aims to determine whether a given example is out-of-distribution (OOD) or in-distribution (ID) by assigning an estimated uncertainty score. An OOD example is labeled as 1, while an ID example is labeled as 0. The experiments of the OOD detection are conducted using the left-out class setting, which aligns with the procedure in Zhao et al. (2020) and Stadler et al. (2021). Note that we exclude the left-out class from the training set but retain the nodes belonging to this class in the graph. Specifically, we randomly select one category as the OOD class, while the remaining categories are considered ID classes. Within each dataset, we create four random configurations, designating one distinct class as OOD in each configuration. The weighted average, factoring in the number of test OOD nodes for every dataset, is displayed in Table 2. Please refer to Appendix E.5 for more detailed settings and results. In Table 2, the bold numbers denote the best results over all types of architectures (i.e. GCN, EGCN, and GPN), while the underlined numbers are the best results within each type if it is not highlighted.

Table 2: AUROC and AUPR for the OOD Detection.

| dataset | UP | | UH | | KSC | |
|---|---|---|---|---|---|---|
| | AUROC | AUPR | AUROC | AUPR | AUROC | AUPR |
| softmax-GCN | 57.04±5.80 | 16.34±3.29 | 56.78±2.63 | 19.18±0.57 | 77.12±0.65 | 54.18±1.29 |
| RGAE | 77.22±n.a. | 24.81±n.a. | 52.51±n.a. | 10.59±n.a. | 69.62±n.a. | 34.10±n.a. |
| TLRSR | 74.03±n.a. | 20.11±n.a. | 48.95±n.a. | 6.24±n.a. | 58.14±n.a. | 9.84±n.a. |
| TRDFTVAD | 68.70±1.28 | 17.72±0.75 | n.a.±n.a. | n.a.±n.a. | 30.47±2.38 | 12.13±0.69 |
| EGCN | 87.21±0.67 | 45.50±0.65 | 88.64±0.33 | 39.68±1.77 | 89.29±0.13 | 70.17±0.55 |
| EGCN - UR | 90.43±0.18 | 46.06±0.31 | 89.81±0.56 | 43.25±2.75 | 89.45±0.32 | 70.81±1.11 |
| EGCN - UR -TV(**Ours**) | 91.57±0.12 | 46.44±0.18 | **90.69**±0.46 | 46.77±2.90 | **92.21**±0.42 | **72.13**±1.63 |
| GPN | 82.82±3.21 | 40.96±2.50 | 82.16±1.25 | 46.30±3.07 | 79.66±3.82 | 59.30±0.59 |
| GPN - UR | 93.63±0.62 | 48.71±1.87 | 84.75±0.76 | 49.57±1.07 | 88.40±0.80 | 62.01±0.81 |
| GPN -UR -TV (**Ours**) | **94.55**±0.23 | **51.84**±0.72 | 87.29±1.04 | 52.02±2.36 | 88.78±1.71 | 63.11±1.11 |

The bold numbers are the best results over all models. The underlined numbers are the best results within the same model type. n.a. means either model or metric not applicable.

We observe that our proposed uncertainty quantification framework with both EGCN and GPN outperforms softmax-GCN and three anomaly detection baselines. First, this suggests that softmax entropy cannot effectively capture the epistemic uncertainty, resulting in poor performance on OOD detection. Second, anomaly detection techniques aimed at identifying pixels with abnormal features struggle to recognize OOD samples, possibly because abnormal features do not consistently align with features identifying OOD samples. In addition, the GPN backbone performs the best on UP, while EGCN performs best on UH and KSC. This difference in performance may be attributed to the GKDE regularization in the EGCN model, which appears to have a stronger influence on UH and KSC, crucial for the OOD detection performance of the EGCN model. As experimental evidence, we utilize the GKDE regularization for OOD detection and achieve ROC values of 87.24% and 87.2% on UH and KSC respectively, while 69.88% on UP.

**Discussions.** We emphasize the contributions of the proposed UR and TV terms based on Table 1 and Table 2. The key findings are as follows. For misclassification detection, both UR and TV demonstrate an ability to enhance the performance of EGCN and GPN. Specifically, UR improves performance in 5 out of 6 cases, while TV improves performance in 3 out of 6 cases. In terms of OOD detection, both UR and TV exhibit significant improvements across all datasets. GPN-UR achieves enhancements of up to 10% in AUROC and 7% in AUPR observed on the UP dataset. Subsequently, we apply TV regularization to EGCN-UR and GPN-UR, showing a 1% increase in AUROC and a 3.1% increase in AUPR on the UP dataset compared to GPN-UR. A comprehensive ablation study is provided in Appendix E.1 and Table 12.

It is also worth noting that AUROC and AUPR values are not always consistent. For example, on UH, GPN-UR-TV has a higher AUPR but a lower AUROC than EGCN-UR-TV. AUROC and AUPR offer different perspectives for measuring the quality of a ranking on data points to separate positives and negatives. Davis & Goadrich (2006) pointed out that algorithms optimized for AUROC are not guaranteed to optimize AUPR, and vice versa. Yuan et al. (2023) also reported similar empirical observations for classification tasks. A low AUROC but a high AUPR for GPN-UR-TV indicates that GPN produces more true positives among the top-ranked nodes than EGCN, while EGCN can separate true positives and negatives better than GPN among the bottom-ranked nodes.

To further illustrate the above observation, Figure 3 displays example curves of AUROC and AUPR on the UP dataset, with "shadows" selected as the OOD class. While TLRSR and RGAE exhibit impressive AUROC performance (over 93%), their AUPR outcomes are notably poor (below 12%), in contrast to the PR of our proposed framework (over 93%). A high AUROC with an extremely low AUPR for a balanced dataset indicates that the model tends to produce tremendous false positive errors. For

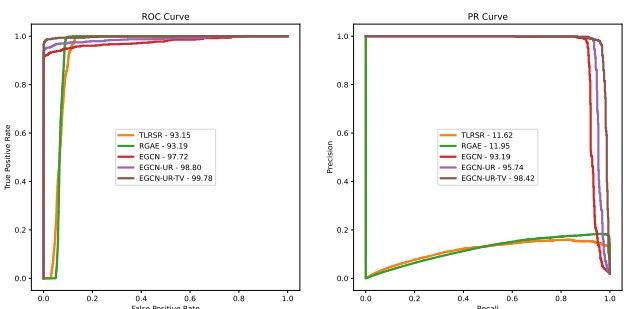

Figure 3: OOD detection on UP with "shadows" as OOD class.

example, most nodes have similar high predicted uncertainty scores, making them indistinguishable between ID and OOD.

## 5    CONCLUSION

We propose a graph-based uncertainty quantification framework for HSIC that is novel in deep learning and hyperspectral literature. We point out scenarios when UCE cannot separate ID and OOD nodes. To mitigate the limitation, we leverage inherent physical characteristics of HS data and edge-preserving regularization to propagate evidence in the spatial domain, leading to unmixing regularization (UR) and evidence-based total variation (TV), respectively. We conduct experiments on three datasets to demonstrate the effectiveness of the proposed regularizations. As the effectiveness of the UR term largely relies on the assumption of the linear mixing model (9), we will develop a more stable HSCI model subject to errors introduced by inaccurate mixing model and mixing matrix (please refer to Appendix G for limitations of the proposed approach). Other future directions include using superpixels to build the graph for the sake of complexity and multiple OOD material categories (as opposed to only one in this work).

ACKNOWLEDGMENTS

We would like to thank Jocelyn Chanussot at the Grenoble Institute of Technology for his helpful discussion on the hyperspectral unmixing problem and benchmark datasets. This work is partially supported by the National Science Foundation (NSF) under Grant No. 2414705, 2220574, 2107449, and 1750911.

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

# A PROOFS FOR THEORETICAL RESULTS

## A.1 PROOFS FOR LIMITATIONS OF UCE

**Proposition 1.** *Suppose $\mathbf{z} \in \mathcal{D} \subset \mathbb{R}^D$ is a point in the embedded space and $y \in \{-1, +1\}$ is its binary class label. An MLP-based ENN has the lower and upper bounds for the UCE loss:*

$$\frac{1}{e_y(\mathbf{z}; \boldsymbol{\theta}) + 1} \leq UCE(\boldsymbol{\alpha}(\mathbf{z}), y; \boldsymbol{\theta}) \leq \frac{\lceil e_{-y}(\mathbf{z}; \boldsymbol{\theta}) \rceil + 1}{e_y(\mathbf{z}; \boldsymbol{\theta})}, \tag{18}$$

*where $e_y(\mathbf{z}; \boldsymbol{\theta})$ refers to the output evidence for the class $y$ and $\boldsymbol{\alpha}(\mathbf{z}) = \mathbf{e}(\mathbf{z}; \boldsymbol{\theta}) + 1$. If $\boldsymbol{\theta}$ can predict $y$ correctly: $y(e_+(\mathbf{z}; \boldsymbol{\theta}) - e_-(\mathbf{z}; \boldsymbol{\theta})) > 0$, we have the following tighter upper bound:*

$$UCE(\boldsymbol{\alpha}(\mathbf{z}), y; \boldsymbol{\theta}) \leq \overline{UCE}(\boldsymbol{\alpha}(\mathbf{z}), y; \boldsymbol{\theta}) := \frac{r + 1}{e_y(\mathbf{z}; \boldsymbol{\theta})}, \tag{19}$$

*where $r = 0$ if the output activation function is ReLU and $r = 1$ if it is the exponential function.*

*Proof.* The UCE loss function has an analytical form:

$$\begin{aligned}
\text{UCE}(\boldsymbol{\alpha}(\mathbf{z}), y; \boldsymbol{\theta}) &= \mathbb{E}_{\boldsymbol{p} \sim \text{Dir}(\boldsymbol{p}|\boldsymbol{\alpha}(\mathbf{z}))} \left[ -\log \mathbb{P}(\mathbf{y}|\boldsymbol{p}) \right] \\
&= \Psi(e_y(\mathbf{z}; \boldsymbol{\theta}) + e_{-y}(\mathbf{z}; \boldsymbol{\theta}) + 2) - \Psi(e_y(\mathbf{z}; \boldsymbol{\theta}) + 1),
\end{aligned}$$

where $\Psi(\cdot)$ is the digamma function and $-y$ refers to a different class label other than $y$. For example, if $y = +1$, then $-y = -1$ refers to the negative class. As $\Psi(\cdot)$ is a monotonic increasing function, we have the lower bound:

$$\begin{aligned}
\text{UCE}(\boldsymbol{\alpha}(\mathbf{z}), y; \boldsymbol{\theta}) &= \Psi(e_y(\mathbf{z}; \boldsymbol{\theta}) + e_{-y}(\mathbf{z}; \boldsymbol{\theta}) + 2) - \Psi(e_y(\mathbf{z}; \boldsymbol{\theta}) + 1) \\
&\geq \Psi(e_y(\mathbf{z}; \boldsymbol{\theta}) + \lfloor e_{-y}(\mathbf{z}; \boldsymbol{\theta}) \rfloor + 2) - \Psi(e_y(\mathbf{z}; \boldsymbol{\theta}) + 1) \\
&\geq \Psi(e_y(\mathbf{z}; \boldsymbol{\theta}) + 2) - \Psi(e_y(\mathbf{z}; \boldsymbol{\theta}) + 1).
\end{aligned}$$

It follows from the recurrence relation of the digamma function, i.e., $\Psi(x+1) = \Psi(x) + 1/x, \forall x > 0$ that a lower bound of the UCE loss function:

$$\text{UCE}(\boldsymbol{\alpha}(\mathbf{z}), \mathbf{y}; \boldsymbol{\theta}) \geq \Psi(e_y(\mathbf{z}; \boldsymbol{\theta}) + 2) - \Psi(e_y(\mathbf{z}; \boldsymbol{\theta}) + 1) = \frac{1}{e_y(\mathbf{z}; \boldsymbol{\theta}) + 1}. \tag{20}$$

Similarly, we can achieve an upper bound:

$$\begin{aligned}
\text{UCE}(\boldsymbol{\alpha}(\mathbf{z}), y; \boldsymbol{\theta}) &= \Psi(e_y(\mathbf{z}; \boldsymbol{\theta}) + e_{-y}(\mathbf{z}; \boldsymbol{\theta}) + 2) - \Psi(e_y(\mathbf{z}; \boldsymbol{\theta}) + 1) \\
&\leq \Psi(\lceil e_{-y}(\mathbf{z}; \boldsymbol{\theta}) \rceil + e_y(\mathbf{z}; \boldsymbol{\theta}) + 2) - \Psi(e_y(\mathbf{z}; \boldsymbol{\theta}) + 1) \\
&= \sum_{i=1}^{\lceil e_{-y}(\mathbf{z}; \boldsymbol{\theta}) \rceil + 1} \frac{1}{e_y(\mathbf{z}; \boldsymbol{\theta}) + i} \leq \frac{\lceil e_{-y}(\mathbf{z}; \boldsymbol{\theta}) \rceil + 1}{e_y(\mathbf{z}; \boldsymbol{\theta})}.
\end{aligned}$$

we get a desired upper bound as in (18).

In the separable case, i.e. $y(e_+(\hat{\mathbf{z}}; \boldsymbol{\theta}) - e_-(\hat{\mathbf{z}}; \boldsymbol{\theta})) > 0$, which implies that the training examples in the training set $\mathcal{D}$, i.e., $(\hat{\mathbf{z}}, y)$ can be correctly classified based on the projected class probabilities: $y(p_+(\hat{\mathbf{z}}) - p_-(\hat{\mathbf{z}})) > 0$, where $[p_+(\hat{\mathbf{z}}), p_-(\hat{\mathbf{z}})] = [(e_+(\hat{\mathbf{z}}; \boldsymbol{\theta}) + 1)/S, (e_-(\hat{\mathbf{z}}; \boldsymbol{\theta}) + 1)/S]$ and $S = e_+(\hat{\mathbf{z}}; \boldsymbol{\theta}) + e_-(\hat{\mathbf{z}}; \boldsymbol{\theta}) + 2$.

The last MLP layer of the ENN can be defined as:

$$\mathbf{e}(\mathbf{z}; \boldsymbol{\theta}) := [e_+(\hat{\mathbf{z}}; \boldsymbol{\theta}), e_-(\hat{\mathbf{z}}; \boldsymbol{\theta})] = [\sigma(\mathbf{w}^T \hat{\mathbf{z}} + b), \sigma(-\mathbf{w}^T \hat{\mathbf{z}} - b)], \tag{21}$$

where $\hat{\mathbf{z}}$ is the input to the last MLP layer $\boldsymbol{\theta} = \{\mathbf{w}, b\}$, $\mathbf{w} \in \mathbb{R}^D$, $b \in \mathbb{R}$, and $\sigma(\cdot)$ is the activation function (e.g. ReLU and exponential) that outputs evidence values. If the configuration $\boldsymbol{\theta}$ can separate the example $(\mathbf{x}, y)$ correctly: $y(e_+(\hat{\mathbf{z}}; \boldsymbol{\theta}) - e_-(\hat{\mathbf{z}}; \boldsymbol{\theta})) > 0$, we have that $e_+(\hat{\mathbf{z}}; \boldsymbol{\theta}) > e_-(\hat{\mathbf{z}}; \boldsymbol{\theta})$ for $y = +1$ and $e_+(\hat{\mathbf{z}}; \boldsymbol{\theta}) < e_-(\hat{\mathbf{z}}; \boldsymbol{\theta})$ for $y = -1$. We discuss two types of activation functions separately.

a) **Exponential function**: $\exp(\mathbf{w}^T \hat{\mathbf{z}} + b) > \exp(-\mathbf{w}^T \hat{\mathbf{z}} - b)$ for $y = +1$ and $\exp(\mathbf{w}^T \hat{\mathbf{z}} + b) < \exp(-\mathbf{w}^T \hat{\mathbf{z}} - b)$ for $y = -1$, which implies that $0 \leq \exp(-\mathbf{w}^T \hat{\mathbf{z}} - b) \leq 1$ for $y = +1$ and $0 \leq \exp(\mathbf{w}^T \hat{\mathbf{z}} + b) \leq 1$ for $y = -1$. It follows that: $\lceil e_{-y}(\hat{\mathbf{z}}, \boldsymbol{\theta}) \rceil \leq 1$. Therefore, we obtain a tighter upper bound in Equation (19) with $r = 1$.

b) **ReLU function**: $\text{ReLU}(\mathbf{w}^T\hat{\mathbf{z}} + b) > 0$ and $\text{ReLU}(-\mathbf{w}^T\hat{\mathbf{z}} - b) = 0$ when $y = +1$. When $y = -1$, we $\text{ReLU}(\mathbf{w}^T\hat{\mathbf{z}} + b) = 0$ and $\text{ReLU}(-\mathbf{w}^T\hat{\mathbf{z}} - b) > 0$. we then have: $\lceil e_{-y}(\hat{\mathbf{z}}, \boldsymbol{\theta}) \rceil = 0$. We obtain a tighter upper bound in Equation (19) with $r = 0$.

$\square$

**Lemma 2.** *Assume that the universal approximation property holds for an MLP-based ENN, i.e., ENN can learn an arbitrary mapping function from the feature vector $\mathbf{z}$ to the evidence values of binary classes: $\mathbf{e}(\mathbf{z};\boldsymbol{\theta}) = [e_+(\mathbf{z};\boldsymbol{\theta}), e_-(\mathbf{z};\boldsymbol{\theta})]^T \in [0,\infty)^2$. Given a training set $\mathcal{D} = \{(\mathbf{z}^i, y^i)\}_{i=1}^N$ and loss function $UCE(\mathcal{D}, \boldsymbol{\theta})$, the model optimization can get a minimal UCE loss for each training sample, i.e., $\boldsymbol{\theta}^\star = \arg\min_{\boldsymbol{\theta}} UCE(\mathbf{z}, y, \boldsymbol{\theta}), \forall (\mathbf{z}, y) \in \mathcal{D}$. Additionally, we have,*

$$\lim_{UCE(\mathbf{z},y,\boldsymbol{\theta})\to 0} e_y(\mathbf{z};\boldsymbol{\theta}) = +\infty \tag{22}$$

$$\lim_{UCE(\mathbf{z},y,\boldsymbol{\theta})\to 0} e_{-y}(\mathbf{z};\boldsymbol{\theta}) = 0. \tag{23}$$

*Proof.* Thanks to the universal approximation property, the optimal solution $\boldsymbol{\theta}^\star$ can predict the Dirichlet distribution $Dir(\boldsymbol{\alpha}^\star)$ that has the minimal UCE loss for each training example $(\mathbf{z}, y) \in \mathcal{D}$. Minimizing $UCE(\mathbf{z}, y, \boldsymbol{\theta})$ over $\boldsymbol{\theta}$ is equivalent to minimizing over $\boldsymbol{\alpha}$, i.e.,

$$\min_{\boldsymbol{\theta}} \text{UCE}(\mathbf{z}, y, \boldsymbol{\theta}) = \min_{\boldsymbol{\alpha}} \mathbb{E}_{\mathbf{p} \in Dir(\boldsymbol{\alpha})}[\ell_{CE}(\mathbf{p}, y)], \tag{24}$$

where $\ell_{CE}(\mathbf{p}, y)$ denotes the standard cross entropy function. As $\ell_{CE}$ is convex, we apply the Jensen's inequality to get,

$$\mathbb{E}_{\mathbf{p} \in Dir(\boldsymbol{\alpha})}[\ell_{CE}(\mathbf{p}, y)] \geq \ell_{CE}(\mathbb{E}_{\mathbf{p} \in Dir(\boldsymbol{\alpha})}[\mathbf{p}], y). \tag{25}$$

Denote $\bar{\mathbf{p}} = \mathbb{E}_{\mathbf{p} \in Dir(\boldsymbol{\alpha})}[\mathbf{p}] = [\bar{p}_+, \bar{p}_-]$. Minimizing the right side of Equation 25, we have $\bar{p}^*_y \to 1$ and $\bar{p}^*_{-y} \to 0$ (Note, we consider that the probability never achieves zero). We also have the analytical expectation of a random variable following the Dirichlet distribution, i.e., $\bar{\mathbf{p}} = \frac{\boldsymbol{\alpha}}{\mathbf{1}^T\boldsymbol{\alpha}} = [\frac{\alpha_+}{\alpha_+ + \alpha_-}, \frac{\alpha_-}{\alpha_+ + \alpha_-}]$. Together with the equation $\boldsymbol{\alpha} = \mathbf{e}(\mathbf{z};\boldsymbol{\theta}) + \mathbf{1}$, we have $e_y(\mathbf{z};\boldsymbol{\theta}^\star) \to +\infty$ and $e_{-y}(\mathbf{z};\boldsymbol{\theta}^\star) \to 0$, as $UCE(\mathbf{z}, y, \boldsymbol{\theta}) \to 0$.

$\square$

**Theorem 1.** *We assume that (i) feature vectors belonging to classes $\{\pm 1\}$ follow Gaussian distributions with the same covariance matrix and the means $\pm\boldsymbol{\mu}$, respectively, i.e., $\mathbb{P}(\mathbf{z}, y) = \mathbb{P}(y = +1)\mathcal{N}(\mathbf{z}; \boldsymbol{\mu}, \boldsymbol{\Sigma}) + \mathbb{P}(y = -1)\mathcal{N}(\mathbf{z}; -\boldsymbol{\mu}, \boldsymbol{\Sigma})$, with $\mathbb{P}(y = +1) = \mathbb{P}(y = -1) = 0.5$; (ii) the optimal solutions $\boldsymbol{\theta}^\star$ that minimize $\mathbb{E}_{(\mathbf{z},y)\sim\mathbb{P}(\mathbf{z},y)}\overline{UCE}(\boldsymbol{\alpha}(\mathbf{z}), \mathbf{y}; \boldsymbol{\theta})$ can separate both classes: $e_y(\mathbf{z};\boldsymbol{\theta}) > e_{-y}(\mathbf{z};\boldsymbol{\theta}), \forall(\mathbf{z}, y)$. Let $\sigma(\cdot)$ be the exponential function. The optimal solution $\boldsymbol{\theta}^\star = (\mathbf{w}^\star, b^\star)$ is the same as the optimal solution of LDA, i.e.,*

$$\mathbf{w}^\star = \boldsymbol{\Sigma}^{-1}\boldsymbol{\mu} \text{ and } b^\star = 0. \tag{26}$$

*Proof.* According to the assumption on Gaussian distributions for the two classes, the upper bound of UCE loss, i.e., $\overline{UCE}$, has the following relations:

$$\mathbb{E}_{(\mathbf{z},y)\sim P(\mathbf{z},y)}\overline{UCE}(\boldsymbol{\alpha}(\mathbf{z}), \mathbf{y}; \boldsymbol{\theta})$$

$$= \frac{1}{2}\mathbb{E}_{\mathbf{z}\sim\mathcal{N}(\boldsymbol{\mu},\boldsymbol{\Sigma})}\overline{UCE}(\boldsymbol{\alpha}(\mathbf{z}), +\mathbf{1}; \boldsymbol{\theta}) + \frac{1}{2}\mathbb{E}_{\mathbf{z}\sim\mathcal{N}(-\boldsymbol{\mu},\boldsymbol{\Sigma})}\overline{UCE}(\boldsymbol{\alpha}(\mathbf{z}), -\mathbf{1}; \boldsymbol{\theta})$$

$$= \mathbb{E}_{\mathbf{z}\sim\mathcal{N}(\boldsymbol{\mu},\boldsymbol{\Sigma})}\left[\frac{1}{e_+(\mathbf{z};\boldsymbol{\theta})}\right] + \mathbb{E}_{\mathbf{z}\sim\mathcal{N}(-\boldsymbol{\mu},\boldsymbol{\Sigma})}\left[\frac{1}{e_-(\mathbf{z};\boldsymbol{\theta})}\right]$$

$$= \mathbb{E}_{\mathbf{z}\sim\mathcal{N}(\boldsymbol{\mu},\boldsymbol{\Sigma})}\exp(-\mathbf{w}^T\mathbf{z} - b) + \mathbb{E}_{\mathbf{z}\sim\mathcal{N}(-\boldsymbol{\mu},\boldsymbol{\Sigma})}\exp(\mathbf{w}^T\mathbf{z} + b). \tag{27}$$

It holds $-\mathbf{w}^T\mathbf{z} - b \sim \mathcal{N}(-\mathbf{w}^T\boldsymbol{\mu} - b, \mathbf{w}^T\boldsymbol{\Sigma}\mathbf{w})$ for $\mathbf{z} \sim \mathcal{N}(\boldsymbol{\mu}, \boldsymbol{\Sigma})$ and $\mathbf{w}^T\mathbf{z} + b \sim \mathcal{N}(-\mathbf{w}^T\boldsymbol{\mu} + b, \mathbf{w}^T\boldsymbol{\Sigma}\mathbf{w})$ for $\mathbf{z} \sim \mathcal{N}(-\boldsymbol{\mu}, \boldsymbol{\Sigma})$. Using $\mathbb{E}[\exp(r)] = \exp(\mu + \sigma^2/2)$ for $r \sim \mathcal{N}(\mu, \sigma^2)$, we get

$$\mathbb{E}_{\mathbf{z}\sim\mathcal{N}(\boldsymbol{\mu},\boldsymbol{\Sigma})}\left[\exp(-\mathbf{w}^T\mathbf{z} - b)\right] = \exp(-\mathbf{w}^T\boldsymbol{\mu} - b + \mathbf{w}^T\boldsymbol{\Sigma}\mathbf{w}/2) \tag{28}$$

$$\mathbb{E}_{\mathbf{z}\sim\mathcal{N}(-\boldsymbol{\mu},\boldsymbol{\Sigma})}\left[\exp(\mathbf{w}^T\mathbf{z} + b)\right] = \exp(-\mathbf{w}^T\boldsymbol{\mu} + b + \mathbf{w}^T\boldsymbol{\Sigma}\mathbf{w}/2). \tag{29}$$

Plugging Equations (28) and (29) into Equation (27) yields

$$
\begin{aligned}
&\mathbb{E}_{(\mathbf{z},y)\sim P(\mathbf{z},y)}\overline{\text{UCE}}(\boldsymbol{\alpha}(\mathbf{z}),\mathbf{y};\boldsymbol{\theta})\\
=\ &\exp(-\mathbf{w}^T\boldsymbol{\mu}-b+\mathbf{w}^T\boldsymbol{\Sigma}\mathbf{w}/2)+\exp(-\mathbf{w}^T\boldsymbol{\mu}+b+\mathbf{w}^T\boldsymbol{\Sigma}\mathbf{w}/2)\\
=\ &\exp(-\mathbf{w}^T\boldsymbol{\mu}+\mathbf{w}^T\boldsymbol{\Sigma}\mathbf{w}/2)(\exp(-b)+\exp(b)).
\end{aligned}
$$

Therefore, the minimization problem of $\overline{\text{UCE}}$ can be expressed as

$$
\begin{aligned}
&\min_{\boldsymbol{\theta}}\mathbb{E}_{(\mathbf{z},y)\sim P(\mathbf{z},y)}\overline{\text{UCE}}(\boldsymbol{\alpha}(\mathbf{z}),\mathbf{y};\boldsymbol{\theta})\\
=\ &\min_{\mathbf{w},b}\exp(-\mathbf{w}^T\boldsymbol{\mu}+\mathbf{w}^T\boldsymbol{\Sigma}\mathbf{w}/2)(\exp(-b)+\exp(b))\\
=\ &\min_{\mathbf{w}}\exp(-\mathbf{w}^T\boldsymbol{\mu}+\mathbf{w}^T\boldsymbol{\Sigma}\mathbf{w}/2)\min_{b}(\exp(-b)+\exp(b))
\end{aligned}
$$

It is straightforward to obtain the optimal solution for $b$, that is,

$$
b^\star = 0 = \arg\min_{b}(\exp(-b)+\exp(b)). \tag{30}
$$

As the exponential function is monotonic, we have an equivalent minimization for $\mathbf{w}$ as follows

$$
\min_{\mathbf{w}}\exp(-\mathbf{w}^T\boldsymbol{\mu}+\mathbf{w}^T\boldsymbol{\Sigma}\mathbf{w}/2)=\min_{\mathbf{w}}-\mathbf{w}^T\boldsymbol{\mu}+\mathbf{w}^T\boldsymbol{\Sigma}\mathbf{w}/2. \tag{31}
$$

By taking the gradient of equation 31 with respect to $\mathbf{w}$ and setting it to zero, we obtain $-\boldsymbol{\mu}+\boldsymbol{\Sigma}\mathbf{w}=0$, which has a closed-form solution, given by

$$
\mathbf{w}^\star = \boldsymbol{\Sigma}^{-1}\boldsymbol{\mu}. \tag{32}
$$

$\square$

## A.2   GRADIENT ANALYSIS OF PROPOSED UNMIXING-BASED REGULARIZATION(UR TERM)

**Proposition 2.** *Assume the linear model (9) holds without noise, the gradient descent for minimizing the UR regularization increases the predicted evidence of ground-truth class for ID instances and decrease the total evidence for OOD instances with the corresponding pure material contained in the pixel; formally, we have*

a) *For an instance $(\boldsymbol{x}^i,y^i)$ with feature matrix $\boldsymbol{x}^i=\boldsymbol{m}_{y^i}$ and $y^i\in\{1,\dots,C\}$ is the ground truth ID class label, one has*

$$
\frac{\partial UR(\boldsymbol{e})}{\partial e^i_{y^i}}\le 0. \tag{33}
$$

b) *For an OOD instance instance $(\boldsymbol{x}^i,y^i)$ with $\boldsymbol{x}^i=\boldsymbol{m}_o^*$ and $y^i=o\notin\{1,\dots,C\}$*

$$
\sum_{c=1}^C\frac{\partial UR(\boldsymbol{e})}{\partial e^i_c}\ge 0. \tag{34}
$$

*Proof.* Given an instance $(\boldsymbol{x}^i,y^i)$ with $\boldsymbol{x}\in\mathbb{R}^B$, $y^i\in\{1,\dots,C,o\}$, ID material signatures $\boldsymbol{m}_c\in\mathbb{R}^B$ for $c=\{1,\dots,C\}$, the OOD material signature $\boldsymbol{m}_o^*\in\mathbb{R}^B$ optimized in Appendix A.3, and the subjective logic opinion $\omega^i=(\boldsymbol{b}^i,u^i)$ is based on model prediction $\boldsymbol{e}^i(\boldsymbol{\theta})$. For brevity, we omit $\boldsymbol{\theta}$ in the rest of the proof, thus getting $\boldsymbol{b}^i=\frac{\boldsymbol{e}^i}{C+\sum_{c=1}^C e^i_c}$, $u=\frac{C}{C+\sum_{c=1}^C e^i_c}$. The UR term can be formulated as

$$
\text{UR}(\boldsymbol{e})=\sum_{i\in\mathbb{V}}\|\boldsymbol{x}^i-\frac{\sum_{c=1}^C e^i_c\boldsymbol{m}_c}{C+\sum_{c=1}^C e^i_c}-\frac{C\boldsymbol{m}_o^*}{C+\sum_{c=1}^C e^i_c}\|_2^2.
$$

Taking the partial derivative to $e^i_k$, which is the evidence scalar of class $k$ for instance $i$, we obtain

$$
\frac{\partial\text{UR}(\boldsymbol{e})}{\partial e^i_k}=-2\left(\boldsymbol{x}^i-\frac{\sum_{c=1}^C e^i_c\boldsymbol{m}_c+C\boldsymbol{m}_o^*}{C+\sum_{c=1}^C e^i_c}\right)^T\frac{(\sum_{c\neq k}e^i_c+C)\boldsymbol{m}_k-\sum_{c\neq k}e^i_c\boldsymbol{m}_c-C\boldsymbol{m}_o^*}{(C+\sum_{c=1}^C e^i_c)^2}. \tag{35}
$$

If $y^i \in \{1, \ldots, C\}$ and $\boldsymbol{x}^i = \boldsymbol{m}_{y^i}$, which indicates the instance is a pure ID material, then we have the partial gradient with respect to the ground truth class $y^i$ as follows:

$$
\begin{aligned}
\frac{\partial \mathrm{UR}(\boldsymbol{e})}{\partial e^i_{y^i}} &= -2 \frac{\left((C + \sum_{c \neq y_i} e^i_c)\boldsymbol{m}_{y^i} - (\sum_{c \neq y^i} e^i_c \boldsymbol{m}_c + C\boldsymbol{m}^*_0)\right)^T \left((\sum_{c \neq y^i} e^i_c + C)\boldsymbol{m}_{y^i} - (\sum_{c \neq y^i} e^i_c \boldsymbol{m}_c + C\boldsymbol{m}^*_0)\right)}{(C + \sum^C_{c=1} e^i_c)^3} \\
&= -2 \frac{\|(C + \sum_{c \neq i} e^i_c)\boldsymbol{m}_i - (\sum_{c \neq y_i} e^i_c \boldsymbol{m}_c + C\boldsymbol{m}^*_0)\|^2_2}{(C + \sum^C_{c=1} e^i_c)^3} \leq 0.
\end{aligned}
\tag{36}
$$

If $y^i = o$ and $\boldsymbol{x}^i = \boldsymbol{m}^*_o$, which implies a pure OOD material, then it further follows from (35) that

$$
\begin{aligned}
\frac{\partial \mathrm{UR}(\boldsymbol{e})}{\partial e^i_k} &= -2 \frac{\left((C + \sum^C_{c=1} e^i_c)\boldsymbol{m}^*_o - (\sum^C_{c=1} e^i_c \boldsymbol{m}_c + C\boldsymbol{m}^*_o)\right)^T \left((\sum_{c \neq k} e^i_c + C)\boldsymbol{m}_k - (\sum_{c \neq k} e^i_c \boldsymbol{m}_c + C\boldsymbol{m}^*_o)\right)}{(C + \sum^C_{c=1} e^i_c)^3} \\
&= \frac{-2\left(\sum^C_{c=1} e^i_c \boldsymbol{m}^*_o - \sum^C_{c=1} e^i_c \boldsymbol{m}_c\right)^T}{(C + \sum^C_{c=1} e_c)^3} \cdot \left((\sum_{c \neq k} e^i_c + C)\boldsymbol{m}_k - (\sum_{c \neq k} e^i_c \boldsymbol{m}_c + C\boldsymbol{m}^*_o)\right).
\end{aligned}
\tag{37}
$$

The gradient descent to update the total evidence can be expressed as

$$
e^i_k := e^i_k - \delta \frac{\partial \mathrm{UR}(\boldsymbol{e})}{\partial e^i_k} \quad \text{and} \quad \sum^C_{k=1} e^i_k := \sum^C_{k=1} e^i_k - \delta \sum^C_{k=1} \frac{\partial \mathrm{UR}(\boldsymbol{e})}{\partial e^i_k},
\tag{38}
$$

where $\delta$ is the learning rate and the summation of class-wise gradient is calculated by

$$
\begin{aligned}
\sum^C_{k=1} \frac{\partial \mathrm{UR}(\boldsymbol{e})}{\partial e^i_k} &= \frac{-2\left(\sum^C_{c=1} e^i_c \boldsymbol{m}^*_o - \sum^C_{c=1} e^i_c \boldsymbol{m}_c\right)^T}{(C + \sum^C_{c=1} e_c)^3} \cdot \left(\sum^C_{k=1}(\sum_{c \neq k} e^i_c + C)\boldsymbol{m}_k - \sum^C_{k=1}(\sum_{c \neq k} e^i_c \boldsymbol{m}_c + C\boldsymbol{m}^*_o)\right) \\
&= \frac{-2\left(\sum^C_{c=1} e^i_c \boldsymbol{m}^*_o - \sum^C_{c=1} e^i_c \boldsymbol{m}_c\right)^T}{(C + \sum^C_{c=1} e_c)^3} \cdot \left(\sum^C_{k=1}(\sum_{c \neq k} e^i_c)\boldsymbol{m}_k + C\sum^C_{c=1} \boldsymbol{m}_c - \sum^C_{k=1}(C - 1)e^i_k \boldsymbol{m}_k - C^2 \boldsymbol{m}^*_o\right) \\
&= \frac{-2\left(\sum^C_{c=1} e^i_c \boldsymbol{m}^*_o - \sum^C_{c=1} e^i_c \boldsymbol{m}_c\right)^T}{(C + \sum^C_{c=1} e_c)^3} \cdot \left(C\sum^C_{c=1}(\boldsymbol{m}_c - \boldsymbol{m}^*_o) + \sum^C_{k=1}(\sum_{c \neq k} e^i_c - (C - 1)e^i_k)\boldsymbol{m}_k\right).
\end{aligned}
$$

Without loss of generality, we assume that there is only one ID class, i.e. $C = 1$, then we have

$$
\sum^C_{k=1} \frac{\partial \mathrm{UR}(\boldsymbol{e})}{\partial e^i_k} = \frac{2C^2(\sum^C_{c=1} e_c)}{(C + \sum^C_{c=1} e_c)^3} \cdot \|\boldsymbol{m}_o - \boldsymbol{m}_{ID}\|^2_2 \geq 0.
$$

$\square$

### A.3 ANALYTICAL SOLUTION FOR SIGNATURE OF OOD MATERIAL

Given the feature set of the whole graph $\{\boldsymbol{x}^i, i \in \mathbb{V}\}$ and each instance has an associated evidence vector $\boldsymbol{e}^i(\boldsymbol{\theta})$ by some fixed $\boldsymbol{\theta}$, the optimal OOD material's signature by minimizing $\mathrm{UR}(\boldsymbol{m}_o)$ over the graph has an analytical form as

$$
\boldsymbol{m}_o = \frac{\sum_{i \in \mathbb{V}}(\boldsymbol{x}^i - \frac{\sum_c e^i_c \boldsymbol{m}_c}{S^i}) \frac{1}{S^i}}{\sum_{i \in \mathbb{V}} \frac{C}{S^{i^2}}}.
\tag{39}
$$

*Proof.* The summation of UR term over the graph can be expressed by

$$
\mathrm{UR}(\boldsymbol{m}_o) = \sum_{i \in \mathbb{V}} \|\boldsymbol{x}^i - \frac{\sum_c e^i_c \boldsymbol{m}_c}{S^i} - \frac{C}{S^i}\boldsymbol{m}_o\|^2_2.
\tag{40}
$$

Taking the derivative of (40) with respect to $\boldsymbol{m}_o$, we have

$$
\begin{aligned}
\frac{\partial \mathrm{UR}(\boldsymbol{m}_o)}{\partial \boldsymbol{m}_o} &= 2 \sum_{i \in \mathbb{V}} (\boldsymbol{x}^i - \frac{\sum_c e^i_c \boldsymbol{m}_c}{S^i} - \frac{C}{S^i}\boldsymbol{m}_o)(-\frac{C}{S^i}) \\
&= -2C \left[\sum_{i \in \mathbb{V}}(\boldsymbol{x}^i - \frac{\sum_c e^i_c \boldsymbol{m}_c}{S^i})\frac{1}{S^i} - \sum_{i \in \mathbb{V}} \frac{C}{S^{i^2}}\boldsymbol{m}_o\right].
\end{aligned}
\tag{41}
$$

Setting $\frac{\partial \mathrm{UR}(\boldsymbol{m}_o)}{\partial \boldsymbol{m}_o} = \boldsymbol{0}$ leads to

$$\boldsymbol{m}_o = \frac{\sum_{i \in \mathbb{V}} (\boldsymbol{x}^i - \frac{\sum_c e_c^i \boldsymbol{m}_c}{S^i}) \frac{1}{S^i}}{\sum_{i \in \mathbb{V}} \frac{C}{S^{i^2}}}. \tag{42}$$

$\square$

## B    DATASET DETAILS

HSI captures data at various wavelengths for a given spatial region. Unlike the human eye, which possesses only three color receptors sensitive to blue, green, and red light, HSI precisely measures a large range of the light spectrum for every pixel in the scene acquired by sensors such as Airborne Visible/Infrared Imaging Spectrometer (AVIRIS) sensor. It offers detailed wavelength resolution not just within the visible range but also in the near-infrared range.

We use three widely used HSI datasets in the HSIC task: the University of Pavia (UP), the University of Houston (UH), and the Kennedy Space Center (KSC). We present details of these three datasets in Table 3. Note that all three datasets used here show high homophily scores Zhu et al. (2020) (exceeding 79% is regarded as a high homophily score), indicating that pixels with similar spectral features tend to belong to the same category.

Table 3: Summary of the three HSI Datasets used for experimental evaluation: "Spatial" is the 2D dimension in terms of width and height; "Spectral" is the spectral bands, i.e., the number of features for each pixel within the "Wavelength" range (nm); "Labeled pixels" counts the labeled pixels in the ground truth datasets and we do not care about the unlabeled pixels; "Training ratio" calculates the proportion of training pixels across all labeled pixels; and the "homophily score" measures how likely nodes with the same label are near each other in a graph.

|  | UP | UH | KSC |
|---|---|---|---|
| Spatial | 610 x 610 | 340 x 1905 | 512 x 614 |
| Spectral | 103 | 144 | 176 |
| Wavelength | 430-860 | 0.35-1.05 | 400-2500 |
| Labeled pixels | 42,776 | 17,270 | 4,364 |
| Categories | 9 | 15 | 13 |
| Training ratio | 8.67% | 27% | 7.72% |
| Homophily score | 0.7913 | 0.7911 | 0.8109 |

The UP dataset was acquired by a Reflective Optics System Imaging Spectrometer (ROSIS) sensor during a flight campaign over the university campus at Pavia, Northern Italy. The detailed class description and training/validation/test ratio are presented in Table 4 following the same setting as (Hong et al., 2020) [2].

Table 4: Land-cover classes of the UP dataset.

| Class No. | Class Name | Training | Validation | Test |
|---|---|---|---|---|
| 0 | Asphalt | 327 | 1260 | 5044 |
| 1 | Meadows | 503 | 3629 | 14517 |
| 2 | Gravel | 284 | 363 | 1452 |
| 3 | Trees | 152 | 582 | 2330 |
| 4 | Painted metal sheets | 232 | 222 | 891 |
| 5 | Bare Soil | 457 | 914 | 3658 |
| 6 | Bitumen | 349 | 196 | 785 |
| 7 | Self-Blocking Bricks | 318 | 672 | 2692 |
| 8 | Shadows | 152 | 159 | 636 |
| | Total | 2774 | 7997 | 32005 |

The UH dataset is collected by the Compact Airborne Spectrographic Imager (CASI) and released as a data fusion contest by The IEEE Geoscience and Remote Sensing Society. Table 5 presents the classes and dataset split following the contest [3].

---

[2]https://github.com/danfenghong/IEEE_TGRS_GCN
[3]http://www.grss-ieee.org/community/technical-committees/data-fusion/2013-ieee-grss-data-fusion-contest/

Table 5: Land-cover classes of the UH dataset.

| Class No. | Class Name | Training | Validation | Test |
|---|---|---|---|---|
| 0 | Healthy grass | 198 | 235 | 941 |
| 1 | Stressed grass | 190 | 252 | 1012 |
| 2 | Artificial turf | 227 | 113 | 455 |
| 3 | Evergreen trees | 188 | 215 | 861 |
| 4 | Deciduous trees | 186 | 222 | 890 |
| 5 | Bare earth | 196 | 28 | 115 |
| 6 | Water | 196 | 256 | 1024 |
| 7 | Residential buildings | 191 | 232 | 931 |
| 8 | Non-residential buildings | 193 | 272 | 1089 |
| 9 | Roads | 191 | 246 | 987 |
| 10 | Sidewalks | 234 | 266 | 1066 |
| 11 | Crosswalks | 192 | 247 | 990 |
| 12 | Major thoroughfares | 246 | 77 | 309 |
| 13 | Highways | 213 | 60 | 240 |
| 14 | Railways | 227 | 114 | 457 |
| | Total | 3068 | 2835 | 11367 |

The KSC dataset was gathered by Airborne Visible/Infrared Imaging Spectrometer (AVIRIS). There is no widely-used public split for KSC and we randomly pick 20 nodes from each class for training following (Kipf & Welling, 2017). The detailed class description and training/validation/test ratio are presented in Table 6.

Table 6: Land-cover classes of the KSC dataset.

| Class No. | Class Name | Training | Validation | Test |
|---|---|---|---|---|
| 0 | Scrub | 20 | 111 | 504 |
| 1 | Willow swamp | 20 | 33 | 152 |
| 2 | Cabbage palm hammock | 20 | 35 | 160 |
| 3 | Cabbage palm/oak hammock | 20 | 34 | 158 |
| 4 | Slash pine | 20 | 21 | 96 |
| 5 | Oak/broadleaf hammock | 20 | 31 | 142 |
| 6 | Hardwood swamp | 20 | 12 | 58 |
| 7 | Graminoid marsh | 20 | 61 | 280 |
| 8 | Spartina marsh | 20 | 75 | 340 |
| 9 | Cattail marsh | 20 | 57 | 261 |
| 10 | Salt marsh | 20 | 59 | 272 |
| 11 | Mud flats | 20 | 72 | 328 |
| 12 | Wate | 20 | 136 | 616 |
| | Total | 260 | 737 | 3367 |

## C GRAPH CONSTRUCTION

Considering most HSI datasets contain a limited number of labeled pixels, we follow a recent work Hong et al. (2020) to build the graph only using these labeled pixels while ignoring the pixels that do not have ground-truth labels. We model the labeled pixels in one HSI scene as a graph $\mathcal{G} = (\mathbb{V}, \mathbb{E})$, where $\mathbb{V} \in \{1, 2, \ldots, N\}$ denotes a vertex set and $\mathbb{E} \subseteq \mathbb{V} \times \mathbb{V}$ is the edge set. Edges can be represented as a weighted adjacency matrix $\boldsymbol{W} \in \mathbb{R}^{N \times N}$ and the weight is calculated with the radial basis function of the similarity between two node features Qin et al. (2020),

$$\boldsymbol{W}_{ij} = \exp^{-d(\boldsymbol{x}^i, \boldsymbol{x}^j)/\sigma},$$

where $d(\boldsymbol{x}^i, \boldsymbol{x}^j)$ is the distance between two vertices $i$ and $j$, such as the Euclidean distance or cosine similarity, and $\sigma > 0$ is a control parameter for the similarity. We use the cosine similarity

$$d(\boldsymbol{x}^i, \boldsymbol{x}^j) = 1 - \frac{<\boldsymbol{x}^i, \boldsymbol{x}^j>}{\|\boldsymbol{x}^i\| \|\boldsymbol{x}^j\|},$$

thanks to its scale invariance, based on the observation that illumination alters the scaling of spectra while preserving their overall shape in the spectral domain (Merkurjev et al., 2014). To improve the computation efficiency with better scalability, we only keep the first $K$ largest weights, considered as nearest neighbor, for each node to build a sparse graph. We choose $K = 50$ and $\sigma = 0.1$ in the experiments.

The built graph exhibits a strong homophily characteristic where nodes typically associate with others that are "similar" or "comparable" from the perspective of their respective categories. It has been shown that GCN manages such highly homophilic graphs effectively (Ma et al., 2021).

## D    MODEL DETAILS

For our comparative analysis, we employ five baseline methods. First is a classification model that utilizes entropy as its uncertainty metric. Additionally, we select three representative anomaly detection techniques, as mentioned in a recent review (Xu et al., 2022). Furthermore, we incorporate two state-of-the-art uncertainty quantification models for semi-supervised node classification.

**Pseudocode for our model**    We provide the pseudo-code for the OOD detection in Algorithm 1.

---

**Algorithm 1** Model Optimization for OOD Detection

---

1: Initialize neural network parameters $\boldsymbol{\theta}^0$ and OOD endmember $\boldsymbol{m}_o^0$ randomly and set $k = 0$.
2: **repeat**
3:     Fix $\boldsymbol{m}_o^k$, then optimize neural network parameters using regularized learning function (Equation 17) and get the $\boldsymbol{\theta}^{k+1}$
4:     Get the prediction of beliefs and epistemic uncertainty (i.e abundance coefficient) with $\boldsymbol{\theta}^{k+1}$
5:     Fix $\boldsymbol{\theta}^{k+1}$, calculate the optimal $\boldsymbol{m}_o^{k+1}$ based on Equation 12
6:     $k = k + 1$
7: **until** convergence
8: Output optimized $\boldsymbol{\theta}^* = \boldsymbol{\theta}^k$ and $\boldsymbol{m}_o^* = \boldsymbol{m}_o^k$.

---

For GCN-based models, we use two graph convolution layers and 0.5 dropout probability. Following the graph size, KSC, UP, and UH have hidden dimensions of 64, 128, 256, respectively. We use early stopping with the patience of 30, a maximum of 5,000 epochs, and validation cross-entropy as a stop metric. For all models, we use the Adam optimizer, and the learning rate and weight decay are carefully tuned for each dataset.

**Softmax-GCN.**    We use classic two-layer GCNs optimized with cross-entropy loss (Hendrycks & Gimpel, 2016) based on the assumption that correctly classified examples tend to have greater maximum softmax probabilities than erroneously classified OOD examples.

Table 7: Hyperparamters for softmax-GCN Model

| dataset | lr | wd |
|---------|----------|----------|
| UP | 1.00E-02 | 1.00E-05 |
| UH | 1.00E-02 | 1.00E-05 |
| KSC | 1.00E-03 | 1.00E-05 |

**TLRSR.**    Tensor Low-Rank and Sparse Representation (TLRSR) model (Wang et al., 2022) used a principal component analysis (PCA) method as one preprocessing step to exact a subset of HSI bands, followed by a tensor low-rank framework to preserve the inherent HSI structure, for extracting the LR background part as the dictionary of TLRSR. Following their parameter tuning, we search $\lambda_1$ and $\lambda_2$ from the set $\{0.001, 0.005, 0.01, 0.05, 0.1, 0.2, 0.3\}$. The hyperparameters used in our experiments are listed in Table 8.

**RGAE.**    Robust Graph AutoEncoder (RGAE) (Fan et al., 2021) is a modified autoencoder framework combined with gradient normalization of each sample to make it more robust to noise and anomalies. Besides, it has a graph regularization term for preserving the local geometric structure of the given high-dimensional data. Three hyperparameters need to be tuned carefully, i.e., (1) Trade-off parameter $\lambda$ that balances the regularization term and the range is set to $\{10^{-4}, 10^{-3}, 10^{-2}, 10^{-1}\}$; (2) Number of superpixels $S$ and the rage is set to $\{50, 100, 150, 300, 500\}$; (3) Dimension of hidden layers $n_{hid}$ and the range is $\{20, 40, 60, 80, 100, 120, 140, 160\}$. We use the validation OOD ROC to pick the top performance models. The hyperparameters used in our experiments are in Table 8.

**TRDFTVAD.** Tensor ring (TR) decomposition with total variation (TV) regularization model (TRDFTVAD) (Feng et al., 2023) is proposed for hyperspectral anomaly detection. This method decomposes hyperspectral imagery data into background and anomaly components, leveraging the low-rank nature of the background across spatial and spectral dimensions, while employing the TV regularization to enhance the piecewise smoothness of the background. Two tuning parameters are $\lambda$ as the weight of TV regularization and $\beta$ as the weight of sparse regularization term. These two parameters are both searched from {0.01, 0.05, 0.1, 0.5, 1, 5,10}. We use the OOD AUROC value on the validation set to select the best performance models. The hyperparameters used in our experiments are in Table 8.

Table 8: Hyperparamters for anomaly detection baselines

| Model | hyper parameters | UP | | | | UH | | | | KSC | | | |
|---|---|---|---|---|---|---|---|---|---|---|---|---|---|
| | | 4 | 6 | 7 | 8 | 0 | 1 | 2 | 10 | 5 | 6 | 7 | 12 |
| RGAE | $\lambda$ | 0.1 | 0.1 | 0.1 | 0.1 | 0.1 | 0.1 | 0.1 | 0.01 | 0.1 | 0.1 | 0.1 | 0.1 |
| | $S$ | 300 | 100 | 100 | 100 | 150 | 150 | 150 | 150 | 500 | 500 | 150 | 150 |
| | $n_{hid}$ | 160 | 120 | 120 | 120 | 80 | 80 | 20 | 40 | 160 | 160 | 40 | 40 |
| TLRSR | $\lambda_1$ | 0.3 | 0.001 | 0.001 | 0.2 | 0.2 | 0.2 | 0.01 | 0.3 | 0.2 | 0.01 | 0.001 | 0.3 |
| | $\lambda_2$ | 0.01 | 0.01 | 0.01 | 0.01 | 0.001 | 0.001 | 0.005 | 0.3 | 0.05 | 0.001 | 0.01 | 0.3 |
| TRDFTVAD | $\lambda$ | 0.01 | 5 | 5 | 10 | n.a. | n.a. | n.a. | n.a. | 0.01 | 0.05 | 10 | 10 |
| | $\beta$ | 0.05 | 0.01 | 0.01 | 0.05 | n.a. | n.a. | n.a. | n.a | 1 | 5 | 0.05 | 0.01 |

**EGCN-GKDE.** Similar to ENN, EGCN uses an activation layer instead of softmax to output non-negative values as the parameters for the predicted Dirichlet distribution. The representation learning step uses GCN layers, while GKDE associated with EGCN is designed to estimate prior Dirichlet distribution parameters for each node, which is calculated based on the shortest path between test nodes and training nodes belonging to different classes on the graph. A necessary condition is that nodes with a high epistemic uncertainty are far away from training nodes and nodes with a high aleatoric uncertainty are near the boundary of classes. It may not exhibit good performance when this condition is not satisfied, i.e., OOD is close to ID training nodes. We present the detailed analysis of GKDE in Section E.2.

In our experiments, we also integrate the GKDE teacher by default. we use $\beta = 0.5$ for KSC and UP, $\beta = 0.2$ for UH. For hyperparameter tuning, we first tune the learning rate (lr) and weight decay (wd), as well as the trade-off parameter for GKDE teacher in $\lambda_2$ based on the average result of ID accuracy and OOD/Misclassification ROC. We suppose there are OOD classes involved in the validation set for hyperparameter selection for all models. The hyperparameters used in our experiments are presented in Table9 and Table 10 for misclassification detection and OOD detection, respectively.

Table 9: Hyperparamters for Misclassification Models

| dataset | GKDE | | | | | GPN | | | | |
|---|---|---|---|---|---|---|---|---|---|---|
| | lr | wd | $\lambda_1$ | $\lambda_2$ | $\lambda_3$ | lr | wd | $\lambda_1$ | $\lambda_2$ | $\lambda_3$ |
| UP | 1.00E-02 | 1.00E-04 | 1.00E-03 | 1.00E-04 | 1.00E-04 | 1.00E-03 | 5.00E-03 | 1.00E-03 | 1.00E-02 | 1.00E-02 |
| UH | 1.00E-02 | 1.00E-04 | 1.00E-02 | 1.00E-04 | 1.00E-05 | 1.00E-04 | 1.00E-04 | 1.00E-04 | 1.00E-01 | 1.00E-04 |
| KSC | 1.00E-02 | 1.00E-04 | 1.00E-03 | 1.00E-02 | 1.00E-04 | 1.00E-02 | 1.00E-03 | 1.00E-03 | 1.00E-04 | 1.00E-02 |

**GPN.** Graph posterior network (GPN) (Stadler et al., 2021) applies multi-layer perceptions for representation learning, followed by normalizing flow for density estimation in the latent space. The graph structure is leveraged for evidence propagation. In detail, GPN consists of three components. (1) A feature encoder $g_\phi$ maps the original node feature $\boldsymbol{x}^i \in \mathbb{R}^B, i \in \mathbb{V}$ onto a low-dimensional latent space $\boldsymbol{z}^i \in \mathbb{R}^H$ with a simple two-layer multi-layer perception (MLP) encoder, $H$ is the latent dimension. i.e. $\boldsymbol{z}^i = g_\phi(\boldsymbol{x}^i)$ and $\phi$ is the encoder parameters. (2) A Radial normalizing flow $h_\varphi$ estimates the density of the latent space per class, which is used to compute the pseudo evidence (class counts) $\beta_c^i := h_\varphi(\boldsymbol{z}^i) = N_c \cdot \mathbb{P}(\boldsymbol{z}^i|c;\varphi)$ (3) A personalized page rank message passing scheme diffuses the pseudo counts (density multiplied by the number of training nodes) by taking the graph structures into account, i.e. $\alpha_c^i = \sum_{v \in \mathbb{V}} \prod_{i,v} \beta_c^v$ with $\prod_{i,v}$ is the dense PPR score reflecting the importance of node $v$ on $i$. Following the GPN paper (Stadler et al., 2021), we

set latent space to have 10 dimensions and 10 radial layers for normalizing flow. We use the CE loss to pre-train the flow layer, teleport is equal to 0.2 and propagation is iterated with 10 steps. For the parameters in the optimizer and tradeoffs for the loss function, the tuning process is the same as EGCN. The parameters used in our experiments are presented in Table9 and Table 10 for misclassification detection and OOD detection, respectively.

Table 10: Hyperparamters for OOD detedction Models

| Dataset | hyper parameters | GPN | | | | | GKDE | | | | |
|---|---|---|---|---|---|---|---|---|---|---|---|
| | | lr | wd | $\lambda_1$ | $\lambda_2$ | $\lambda_3$ | lr | wd | $\lambda_1$ | $\lambda_2$ | $\lambda_3$ |
| UP | 4 | 1.00E-04 | 5.00E-03 | 1.00E-04 | 1.00E-03 | 1.00E-03 | 1.00E-02 | 1.00E-05 | 1.00E-01 | 1.00E-04 | 1.00E-05 |
| | 6 | 1.00E-03 | 5.00E-04 | 1.00E-05 | 1.00E+00 | 1.00E-04 | 1.00E-02 | 1.00E-05 | 0.00E+00 | 1.00E-01 | 1.00E-05 |
| | 7 | 1.00E-03 | 1.00E-03 | 1.00E-03 | 1.00E-01 | 1.00E-05 | 1.00E-03 | 1.00E-05 | 0.00E+00 | 1.00E-02 | 1.00E-05 |
| | 8 | 1.00E-04 | 1.00E-04 | 1.00E-05 | 1.00E-01 | 1.00E-05 | 1.00E-03 | 1.00E-05 | 1.00E-02 | 1.00E-04 | 1.00E-05 |
| UH | 0 | 1.00E-03 | 5.00E-03 | 1.00E-03 | 1.00E-05 | 1.00E-05 | 1.00E-02 | 1.00E-05 | 1.00E-02 | 1.00E-03 | 1.00E-04 |
| | 1 | 1.00E-03 | 1.00E-03 | 1.00E-05 | 1.00E-05 | 1.00E-04 | 1.00E-02 | 1.00E-05 | 1.00E-02 | 1.00E-04 | 1.00E-05 |
| | 2 | 1.00E-04 | 1.00E-04 | 1.00E-04 | 1.00E+00 | 1.00E-03 | 1.00E-02 | 1.00E-05 | 1.00E-02 | 1.00E-04 | 1.00E-04 |
| | 10 | 1.00E-03 | 5.00E-04 | 1.00E-03 | 1.00E-01 | 1.00E-02 | 1.00E-02 | 1.00E-05 | 1.00E-04 | 1.00E-01 | 1.00E-05 |
| KSC | 5 | 1.00E-03 | 5.00E-03 | 1.00E-04 | 1.00E-04 | 1.00E-03 | 1.00E-03 | 1.00E-05 | 1.00E-02 | 1.00E-04 | 1.00E-03 |
| | 6 | 1.00E-03 | 5.00E-03 | 1.00E-04 | 1.00E+00 | 1.00E-03 | 1.00E-03 | 1.00E-05 | 1.00E-02 | 1.00E-01 | 1.00E-02 |
| | 7 | 1.00E-03 | 5.00E-03 | 1.00E-04 | 1.00E+00 | 1.00E-05 | 1.00E-03 | 1.00E-05 | 1.00E-02 | 1.00E-02 | 1.00E-05 |
| | 12 | 1.00E-03 | 5.00E-03 | 1.00E-04 | 1.00E-05 | 1.00E-01 | 1.00E-03 | 1.00E-05 | 1.00E-02 | 1.00E-04 | 1.00E-05 |

**Complexity analysis**  We provide the complexity analysis in Table 11. The three anomaly detection methods (RGAE, TLRSR, TRDFTVAD) are implemented with Matlab and run on a desktop with Intel Core I7-9700 and 16GB memory. The remaining three (softmax-GCN, EGCN-based, GPN-based) are implemented with PyTorch and tested on a single GPU RTX4090 located on a server with AMD Ryzen Threadripper PRO 5955WX and 256GB memory.

Table 11: Complexity Analysis

| | Runing Time (second) | | | Number of parameters | | |
|---|---|---|---|---|---|---|
| | UP | UH | KSC | UP | UH | KSC |
| RGAE | 1,660 | 11,918 | 3,899 | n.a | n.a | n.a |
| TLRSR | 40 | 89 | 91 | n.a | n.a | n.a |
| TRDFTVAD | 2,100 | n.a. | 10,740 | n.a | n.a | n.a |
| softmax-GCN | 177 | 30 | 126 | 14.9k | 41.8k | 24.9k |
| EGCN-based | 320 | 43 | 129 | 14.8k | 41.5k | 24.8k |
| GPN-based | 100 | 80 | 77 | 16.4k | 42.9k | 26.7k |

n.a means either metric is not applicable or model is not tested due the the time limit

# E ADDITIONAL RESULTS

## E.1 ABLATION STUDY

We conduct an ablation study in Table 12 in order to demonstrate the influence of each regularization term applied to EGCN and GPN for OOD detection. The comparison includes the original regularization term used in EGCN/GPN as well as our proposed UR and TV terms, namely $R, UR, TV$ respectively in Equation 17. The EGCN framework inherently incorporates a GKDE teacher (denoted as AT in Table 12) and leverages node-level distance metrics for OOD discernment. The GPN framework applies an entropy regularization (denoted as ER in the table) that encourages a non-peak prediction. We use the UP dataset for the ablation study. Given the results presented in Table 13 and Table 14 show near-ideal performance for scenarios 'UP-4' and 'UP-8', we consider 'Up-6' and 'Up-7' for ablation study, where classes 6 and 7, respectively, are chosen as OOD, with the remainder being treated as in-distribution (ID).

The key findings obtained from the experiment are as follows: (1) GKDE (AT) regularization yields some improvements over UCE-only for 'UP-6' and shows comparable results for 'UP-7'. It indicates

that the GKDE teacher's prior knowledge may not be always true, i.e. OOD test nodes are far away in terms of graph-based distance from the training (ID) nodes compared to ID test nodes (2) The proposed UR term significantly improves the accuracy of OOD detection compared to other baseline models. Specifically, in the context of UP-6 , the UR term augments ROC by 15.97% and 50.79%, and PR by 6.44% and 8.72% for EGCN and GPN respectively. For UP-7, improvements of 6.15% and 16% in ROC together with 4.7% and 16.03% in PR are observed for EGCN and GPN, respectively. (3) The TV regularization term also enhances OOD detection in overall, with improvements ranging from 1.25% to 21.68% in terms of ROC and 1.01% to 4.18% in terms of PR. There is a slight decrease within 1% on EGCN for 'UP-7' and it may be due to reasonable randomness. (3) Models employing both TV and UR terms generally deliver the best results, except for the EGCN model on UP-6. Although the EGCN-TV-UR performs slightly lower (within 1%) ROC and PR than the EGCN-UR, the former achieves better ID OA with 3.05% compared to the latter.

Table 12: Ablation study of regularization terms

| | | | UP- 6 | | | UP-7 | | |
|---|---|---|---|---|---|---|---|---|
| | | | ID OA | ROC | PR | ID OA | ROC | PR |
| EGCN | Baseline | UCE-only | 77.16±0.62 | 72.66±1.10 | 4.08±0.15 | 75.03±3.08 | 80.70±1.87 | 21.73±2.88 |
| | | UCE-AT (EGCN) | 74.86±2.37 | 74.82±4.45 | 4.50±0.67 | **75.91**±2.23 | 80.51±2.13 | 21.72±3.56 |
| | Ours | EGCN-UR | 74.32±1.19 | **90.79**±0.42 | 10.94±0.40 | 72.31±1.70 | 86.66±1.71 | 26.42±3.65 |
| | | EGCN-TV | 77.13±2.15 | 83.96±0.74 | 6.63±0.28 | 75.50±1.24 | 79.81±2.98 | 20.88±4.06 |
| | | EGCN-TV-UR | **77.37**±1.78 | 89.03±1.02 | 9.33±0.74 | 75.05±3.17 | 86.78±1.48 | 26.01±3.32 |
| GPN | Baseline | UCE | 66.92±3.95 | 30.60±7.65 | 1.69±0.15 | 64.36±5.82 | 71.58±8.50 | 14.16±3.94 |
| | | UCE-ER (GPN) | 66.11±7.32 | 39.40±16.80 | 2.06±0.63 | 64.94±3.72 | 74.58±12.10 | 16.51±5.07 |
| | Ours | GPN-UR | 54.67±2.44 | 90.19±0.46 | 10.78±0.34 | 58.73±3.71 | 90.58±1.97 | 32.54±5.22 |
| | | GPN-TV | 66.50±1.56 | 61.08±4.53 | 3.07±0.41 | 61.00±3.31 | 82.33±1.58 | 19.52±0.98 |
| | | GPN-TV-UR | 54.73±1.39 | 90.39±0.34 | **11.31**±0.37 | 66.08±4.39 | **92.89**±0.17 | **42.47**±0.92 |

Bold numbers indicate the best model across all the uncertainty metrics for each dataset.

## E.2 ANALYSIS ON GKDE TEACHER FOR OOD DETECTION

GKDE assumes that OOD test nodes are far away in terms of graph-based distance from the training (ID) nodes compared to ID test nodes. This assumption is not always true in practice. We provide the GKDE form as follows:

$$h_c(y_i, d_{ij}) = \begin{cases} 0 & y^i \neq c \\ g(d_{ij}) & y^i = c, \end{cases}$$

with $g(d_{ij}) = \exp(-\frac{d_{ij}^2}{2\beta^2})$ and $d_{ij}$ denoting the graph distance. The evidence prior $\hat{e}_j = \sum_{i \in \mathbb{L}} \mathbf{h}(y^i, d_{ij})$ with $\mathbf{h}(y^i, d_{ij}) = [h_1, \ldots, h_c, \ldots, h_C]$. $\beta$ is the bandwidth in the Gaussian kernel function.

We investigate the influence of parameter $\beta$ in the GKDE teacher on the OOD detection performance. On a specific OOD setting, pixels belonging to "shadows" in the UP dataset are considered as OOD. Figure 4 shows the OOD detection performance for GKDE prior model with different $\beta$. With $\beta = 1$, GKDE can achieve 96.40% ROC and 70.13% PR value. $\beta = 0.1$ is random ranking result while $\beta = 10$ is a much worse result.

Figure 5 presents the predicted vacuity maps obtained by a variety set of $\beta$ values, in comparison to the ground truth map where OOD is labeled as 1, and ID is 0. Note that unlabeled data in the hyperspectral image is marked as 0 for all sub-figures (including both ground truth and prediction figures). The choice of $\beta$ can significantly influence OOD detection performance. Within a single sub-figure, a comparative analysis reveals a clearer distinction between OOD and ID regions when $\beta = 1$. For $\beta = 0.1$, nearly all pixels exhibit high vacuity, whereas for $\beta = 10$, they display low vacuity uniformly. Hence, adjusting $\beta$ is crucial for accessing a good GKDE prior.

In comparison to the GKDE model, which achieves 99.1% on ROC and 96.9% on PR, the GKDE prior does not perform as effectively as EGCN. As seen in Table 13 through 18, the performance disparity between GKDE and EGCN can reach up to 27% on ROC and 19% on PR. This suggests that the GKDE teachers may not yield optimal results. Conversely, in certain situations, GKDE can indeed enhance the learning process of EGCN.

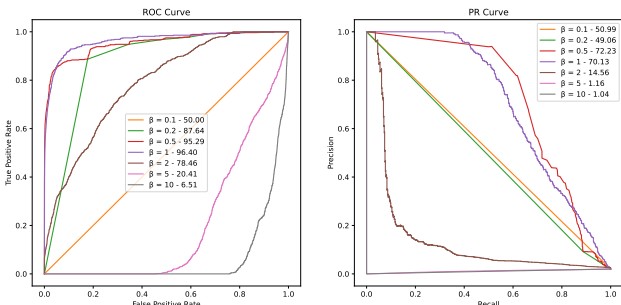

Figure 4: AUROC and AUPR curves for different $\beta$ of GKDE teacher in OOD detection.

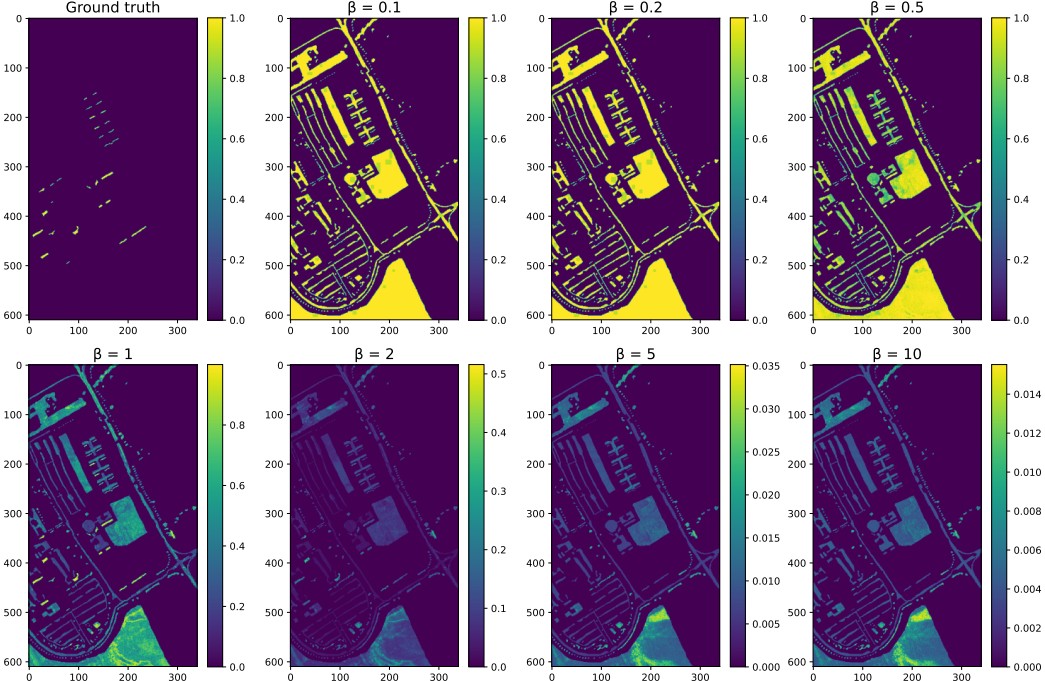

Figure 5: Predicted vacuity maps using different $\beta$ values in the GKDE teacher for OOD detection. The yellow pixels marked in the ground truth subfigure is the OOD pixels and the model is expected to predict high uncertainty for those pixels than the remaining pixels in the picture. $\beta = 1$ has the best ability to distinguish OOD pixels from ID pixels.

## E.3 PARAMETER SENSITIVITY ANALYSIS

Besides the general neural network parameters like learning rate and weight decay, our model incorporates three specific hyperparameters denoted as $\lambda_1, \lambda_2, \lambda_3$ as presented in Equation 17. The parameter $\lambda_1$ is a trade-off weight for the uncertainty quantification framework, i.e. GKDE teacher for EGCN or entropy regularization for GPN. The other two hyperparameters $\lambda_2$ and $\lambda_3$ correspond to the weights for the proposed UR and TV terms, respectively. We conduct sensitivity analysis on these hyperparameters using the variations of $\lambda_2, \lambda_3$ and their impact on the ROC performance. Figure 6 shows that the GPN model is less sensitive to these parameters compared to EGCN. For instance, in the case of 'UP-8', the ROC metric shows stability to variations in $\lambda_2$ when $\lambda_3$ lies within the range of [1e-5, 1e-2]. Similarly for 'Up-7', the ROC metric remains consistent with changes in $\lambda_2$, when $\lambda_3$ is between [0.1, 1]. When the TV term's weight is set small, such as 1e-5, the model becomes highly sensitive to the UR term's weight. Within the EGCN framework, the model's performance is nearly unaffected by $\lambda_3$ when $\lambda_2$ ranges from [1e-5, 1e-4] for 'Up-8', but it decreases significantly when $\lambda_2$ exceeds 1e-4.

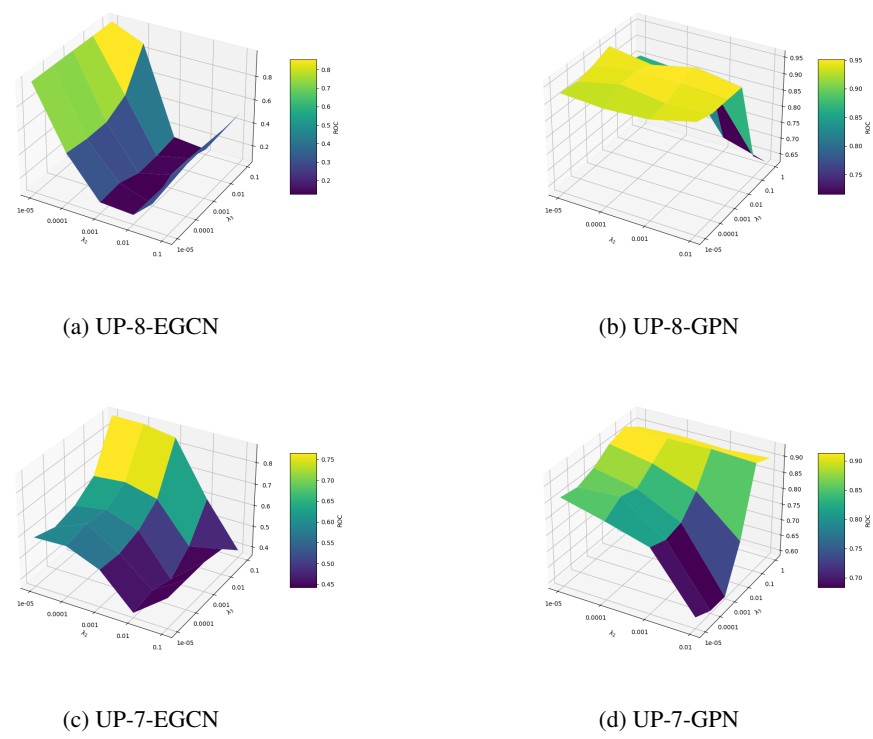

(a) UP-8-EGCN          (b) UP-8-GPN

(c) UP-7-EGCN          (d) UP-7-GPN

Figure 6: Sensitivity analysis of parameters $(\lambda_2, \lambda_3)$ on AUROC for OOD detection.

## E.4 RATIONALS OF THE PROPOSED REGULARIZATIONS

**Explanation of UR performance** Equation (10) assumes that the feature vector corresponding to an ID node is a linear combination of ID endmembers; and consequently, if a feature vector $\mathbf{x}$ is orthonormal to the space that is spanned by $M$, then it is indicative of an OOD node. (10) holds particularly well in scenarios where the OOD spectral signature is orthogonal to $M$. If the OOD spectral is nearly parallel to one of the ID endmembers, then the UR term can not distinguish the OOD from this ID material.

We take the 'UP-6' and 'KSC-5' as two illustrative examples. From Table 15 and 17, we note that the UR term yields an improvement of 18% and 40.4% in ROC for 'UP-6', whereas the improvement for 'KSC-5' is relatively modest, at -0.2% and 4.4% for the EGCN and GPN frameworks respectively. The difference in UR's improvement lies in the difficulty of identifying the OOD class. We present the distribution of cosine distances between OOD nodes and ID nodes of UP-6 and KSC-5 in Figure 7. For 'UP-6', class 'Bituman' (class 6) displays a relatively low degree of similarity (approximately 0.8) compared to nearly 0.9 in KSC-5. This verifies that the UR-based regularization modeled in Equation (10) can improve OOD detection performance under the assumption that OOD spectral should be as orthogonal to the ID feature space as possible.

We also emphasize that the distinctiveness of raw features between ID and OOD data does not inherently ensure the successful detection of OOD nodes when employing the UCE loss. Elaborations on this subject are provided in Section 3.1. To summarize, the UCE loss formulation only involves ID nodes, while disregarding the OOD nodes in the latent space during the feature learning step. Consequently, features that are characteristic of OOD may be overlooked due to the model's focus on ID feature separation. Furthermore, despite potential separability in the latent space, the deterministic boundary delineated by EGCN can erroneously allocate OOD instances to ID regions in the final layer of dimension reduction. Therefore, the property that UR term designed for physical relation between raw features can partially mitigate some drawbacks of the UCE loss.

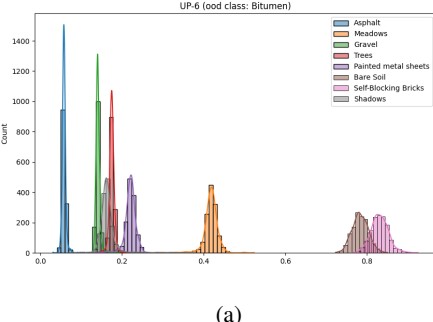 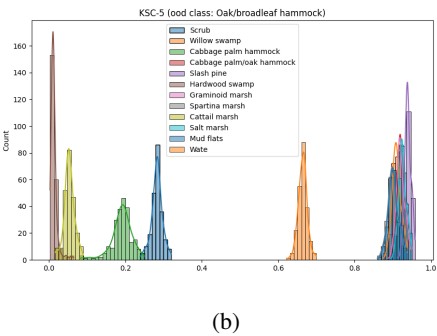

(a)  (b)

Figure 7: The distribution of cosine similarity of the OOD material with all ID endmembers. Each color represents a distinct ID material. The x-axis quantifies the cosine similarity, where smaller values imply less correlation of ID and OOD. If the cosine similarity is closer to 1, it suggests a high degree of similarity, which indicates difficulty in separating ID and OOD.

**Explanation of TV performance**   Comparing 'UH-2' and 'UH-10' in Table 14, the TV term brings significant improvements on 'UH-2'. This is because the TV term is trying to smooth the predicted total evidence (inverse of epistemic uncertainty) across the spatial neighbors and it may bring more gain when the spatial neighbor tends to have similar ground truth evidence. In Figure 8, class 2 ( 'artificial turf') is more concentrated than the class 10 ('sidewalks').

When pixels of a particular class are not densely concentrated, the inclusion of the TV term may not yield significant improvements in out-of-distribution (OOD) detection but rather produce comparable results. There may be several reasons why the combination of TV and UR terms leads to slightly lower OOD detection performance than the UR term alone. For example, there often exists a trade-off between OOD detection performance and in-distribution (ID) classification performance, as discussed in a recent work (Teney et al., 2022). In the case of 'EGCN-UR-TV' applied to the 'Houston-1' dataset, we observe a lower ROC but a higher ID Overall Accuracy (OA). In Table 15, 'KSC-5' demonstrates that 'GPN-UR-TV' has a lower ROC but a higher PR value. Examining the combination of these two metrics provides a more comprehensive perspective, especially for imbalanced scenarios. For 'UP-6,' all three metrics are lower for 'EGCN-UR-TV'. This discrepancy could arise from model randomness, hyperparameter variations, and the performance differences is within a reasonable range.

**Performance difference between EGCN and GPN**   We would like to highlight the distinctions between the two frameworks, EGCN and GPN, as they may exhibit divergent performance patterns.

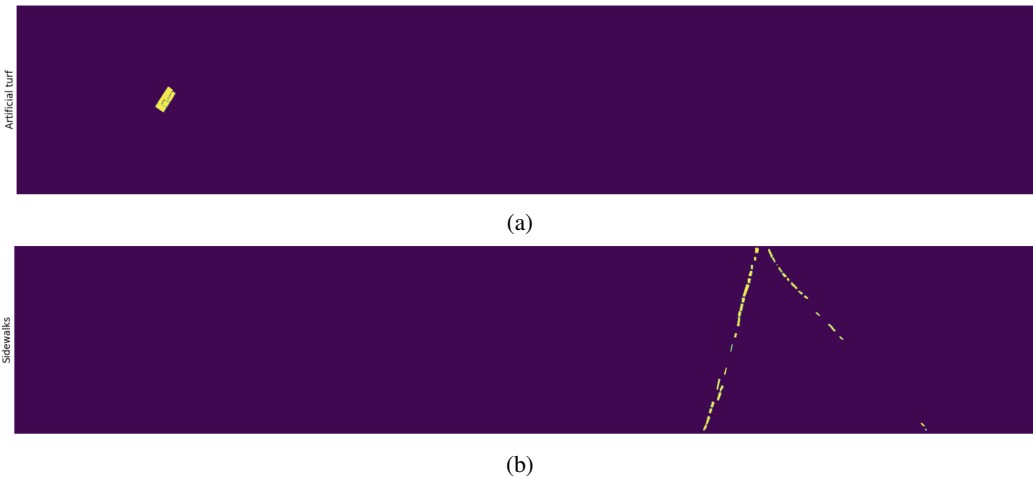

(a)

(b)

Figure 8: Ground truth pixels belonging to class 2 in (a) and class 10 in (b). The pixels are more clustered and concentrated in class 2 compared to class 10, and as a result, TV is more effective (with more improvements) in setting aside class 2 as an OOD class, compared to class 10 as OOD, as shown in Table 16.

Despite both frameworks utilizing the same UCE loss function, EGCN employs a discriminative boundary for prediction, whereas GPN employs a generative model for density estimation. Consequently, these frameworks encounter distinct challenges during the training phase.

### E.5 DETAILED RESULT

For each dataset, we create four random configurations and select one class as OOD. In the main paper, we display the weighted average, factoring in the count of test OOD nodes for every dataset. In this section, we show all the results for the twelve configurations in total.

Tables 13-14 display the results for UP. It is worth noting that some classes are easier to classify while some are challenging. For example, class-4 is easily discernible by both anomaly detection and uncertainty quantification methods. However, the softmax-GCN struggles in this regard, whereas our introduced framework significantly outperforms the anomaly detection benchmark. This may indicate that class-4 possesses distinct features compared to other pixels in the image. On the other hand, for class-6, the PR values are suboptimal across all models, indicating that a considerable number of ID pixels have elevated predicted vacuities. When comparing class-7 and class-8 as the OOD class, models based on GPN fare better with class-7, while those based on EGCN excel with class-8. A similar trend is observed with the GKDE teacher, whucg performs commendably with class-8 but falls short with class-7.

Figure 9 presents the predicted vacuity map for different models when "shadows" (class-8) as the OOD class in UP. TLRSR and RGAE can not identify the OOD at all. EGCN tends to predict higher vacuity scores for OOD nodes (exceeding 0.8) while GKDE has a much lower vacuity score for all nodes (below 0.001). After applying the UR term, there seem to be fewer false positives and lower vacuity scores for ID nodes after applying the TV term.

We have a similar conclusion for the other two datasets. Tables 15-16 present the result for the UH dataset, while Tables 17-18 show the result for KSC. Take the KSC dataset for an example. We observe significant improvements in OOD detection performance for our randomly picked four OOD settings from Table 15 and Table 16 (We ignore the KSC-12 scenario due to the perfect OOD detection performance). Specifically, the inclusion of UR led to a modest improvement of 0.3% - 2.2% on ROC and up to 2.5% on PR for the EGCN model, while the GPN model exhibited remarkable improvements of 4.4%-60.1% on ROC and up to 9.2% on PR. Furthermore, the incorporation of TV resulted in substantial enhancements for both models. The EGCN-UR-TV model demonstrated a notable increase of up to 22.4% on ROC and 5% on PR. The GPN-UR-TV model exhibited a substantial boost of up to 3.7% on ROC and 12.2% on PR. Our findings indicate that the inclusion

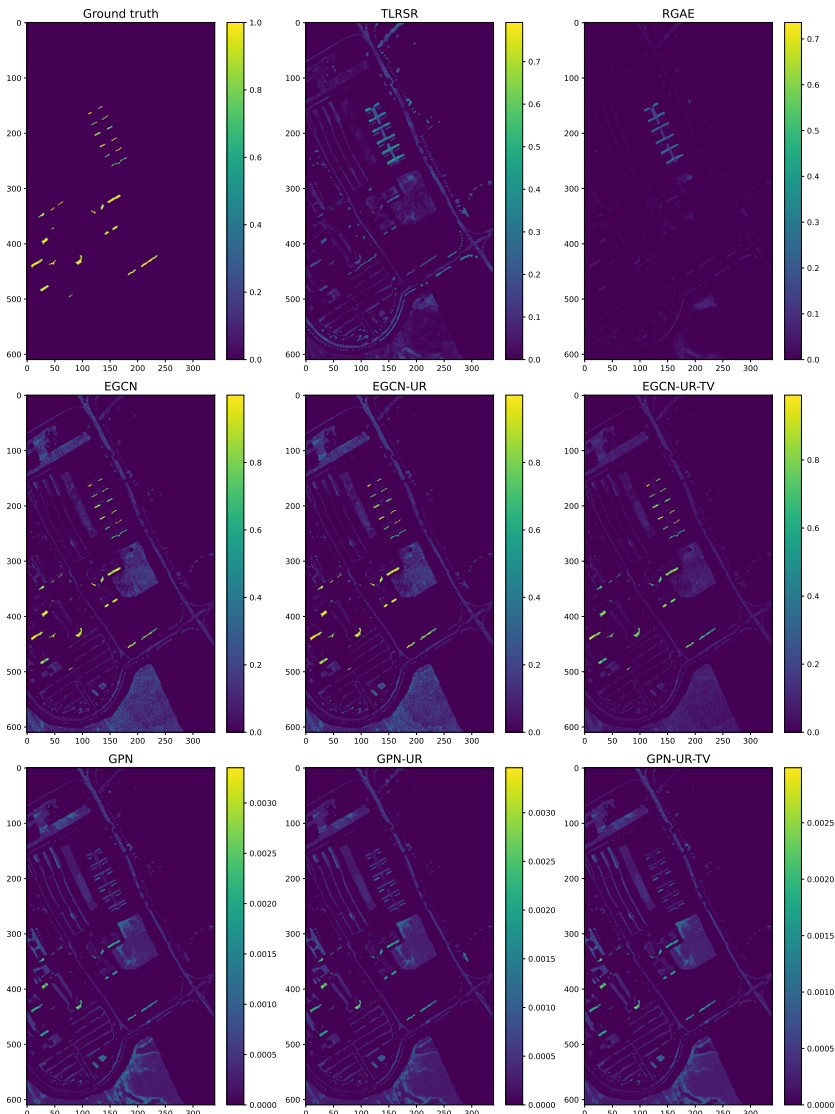

Figure 9: Predicted vacuity map for various competing methods.

of UR and TV terms consistently enhances OOD detection performance on the KSC dataset under the specified experimental settings.

### E.6 ARCHITECTURE ANALYSIS

To investigate the impact of different encoder backbones on the OOD detection, we apply a recent deep GNN architecture (Chen et al., 2020) as the backbone for the EGCN framework, termed EGCN2. The results, as shown in Table 19, align with those obtained using GCN as the encoder, where the UR term notably enhances the ROC and PR metrics for the OOD detection. While the TV term yields comparable results of the OOD detection, it demonstrates improved ID classification performance. Furthermore, a comparison between Tables 12 and Table 19 reveals that GCN2 exhibits superior OOD detection on the 'UP-7' case but shows reduced effectiveness on 'UP-6'.

Table 13: OOD Detection Result for UP

| dataset | UP - 4 | | | UP - 6 | | |
|---|---|---|---|---|---|---|
| | ID OA | OOD ROC | OOD PR | ID OA | OOD ROC | OOD PR |
| softmax-GCN | 74.1±1.8 | 62.9±14.6 | 6.5±6.1 | **77.4**±2.0 | 29.8±2.7 | 1.9±0.1 |
| GKDE | 68.5±n.a. | 99.6±n.a. | 96.8±n.a. | 71.0±n.a. | 59.8±n.a. | 3.1±n.a. |
| RGAE | n.a.±n.a. | 99.3±n.a. | 86.2±n.a. | n.a.±n.a. | 60.6±n.a. | 2.9±n.a. |
| TLRSR | n.a.±n.a. | 98.5±n.a. | 71.4±n.a. | n.a.±n.a. | 70.9±n.a. | 3.8±n.a. |
| TRDFTVAD | n.a.±n.a. | 98.3± 0.1 | 61.34± 2.3 | n.a.±n.a. | 52.9±2.4 | 2.3±0.1 |
| EGCN | 75.5±1.9 | 99.9±0.0 | 97.8±0.3 | 77.0±2.4 | 72.3±0.9 | 4.1±0.1 |
| EGCN - UR | 74.6±1.2 | 99.9±0.0 | 97.1±0.8 | 76.3±0.7 | 90.3±0.2 | 10.4±0.1 |
| EGCN - UR -TV | **76.6**±3.1 | **100.0**±0.0 | **99.8**±0.0 | 75.8±2.2 | 90.1±0.2 | 10.3±0.2 |
| GPN | 71.5±0.0 | 99.6±0.0 | 99.1±0.0 | 69.5±2.7 | 40.4±15.4 | 2.1±0.8 |
| GPN - UR | 71.5±0.1 | 99.6±0.0 | 99.1±0.0 | 50.9±1.7 | 89.9±1.4 | 10.5±1.4 |
| GPN -UR -TV | 63.1±0.6 | 99.6±0.0 | 98.9±0.1 | 51.7±2.0 | 90.7±0.6 | **11.3**±0.8 |

Bold numbers indicate the best model across all the uncertainty metrics for each dataset. Underlined numbers are the best results within the same model type for each dataset. n.a. means either model or metric not applicable. 'UP-4' takes class 4 as the OOD class and 'UP-6' takes class 6 as the OOD class.

Table 14: OOD Detection Result for UP (cont.)

| dataset | UP - 7 | | | UP - 8 | | |
|---|---|---|---|---|---|---|
| | ID OA | OOD ROC | OOD PR | ID OA | OOD ROC | OOD PR |
| softmax-GCN | **76.6**±2.6 | 73.2±3.9 | 27.4±4.0 | **75.3**±1.5 | 14.2±5.5 | 1.3±0.5 |
| GKDE | 70.7±n.a. | 57.0±n.a. | 9.8±n.a. | 68.7±n.a. | 95.3±n.a. | 72.2±n.a. |
| RGAE | n.a.±n.a. | 70.3±n.a. | 12.3±n.a. | n.a.±n.a. | 96.1±n.a. | 18.9±n.a. |
| TLRSR | n.a.±n.a. | 62.3±n.a. | 9.9±n.a. | n.a.±n.a. | 93.1±n.a. | 11.6±n.a. |
| TRDFTVAD | n.a.±n.a. | 57.9± 1.4 | 9.1± 0.3 | n.a.±n.a. | 92.5±1.1 | 12.3±1.3 |
| EGCN | 75.6±0.8 | 84.5±0.9 | 28.1±1.0 | 74.3±1.1 | 99.1±0.2 | 96.9±0.4 |
| EGCN - UR | 75.7±0.8 | 85.2±0.2 | 27.5±0.2 | 73.9±1.2 | 99.3±0.3 | 97.3±0.4 |
| EGCN - UR -TV | 73.0±0.5 | 87.3±0.1 | 26.9±0.2 | 75.0±0.3 | **99.8**±0.0 | **99.1**±0.1 |
| GPN | 65.2±2.6 | 86.5±1.3 | 26.3±1.9 | 68.1±0.3 | 96.0±0.6 | 69.6±10.8 |
| GPN - UR | 62.8±3.3 | 92.2±0.6 | 36.9±2.1 | 67.9±0.5 | 96.0±0.7 | 75.1±4.2 |
| GPN -UR -TV | 62.6±1.9 | **93.2**±0.2 | **40.9**±0.8 | 60.6±2.5 | 97.8±0.1 | 82.5±1.1 |

Bold numbers indicate the best model across all the uncertainty metrics for each dataset. Underlined numbers are the best results within the same model type for each dataset. n.a. means either model or metric not applicable. 'UP-7' takes class 4 as the OOD class and 'UP-8' takes class 6 as the OOD class.

Table 15: OOD Detection Result for UH

| dataset | Houston - 0 | | | Hosuton - 1 | | |
|---|---|---|---|---|---|---|
| | ID OA | OOD ROC | OOD PR | ID OA | OOD ROC | OOD PR |
| softmax-GCN | 66.8±2.9 | 81.9±2.0 | 46.4±0.9 | 67.7±2.2 | 13.8±3.5 | 5.0±0.2 |
| GKDE | 68.9±n.a. | 93.6±n.a. | 68.7±n.a. | 68.6±n.a. | 92.3±n.a. | 66.5±n.a. |
| RGAE | n.a.±n.a. | 85.6±n.a. | 21.6±n.a. | n.a.±n.a. | 46.0±n.a. | 7.7±n.a. |
| TLRSR | n.a.±n.a. | 48.9±n.a. | 6.3±n.a. | n.a.±n.a. | 49.6±n.a. | 8.0±n.a. |
| EGCN | 69.5±0.5 | 93.5±0.4 | 43.8±1.5 | 70.2±2.3 | 96.6±0.2 | 70.1±4.3 |
| EGCN - UR | **71.7**±1.0 | 94.4±0.1 | 47.5±0.8 | 70.3±1.9 | 97.1±0.3 | 73.1±5.0 |
| EGCN - UR -TV | 71.3±1.2 | 96.0±0.7 | 54.2±6.2 | **70.5**±1.0 | 97.0±0.4 | 70.9±1.9 |
| GPN | 66.7±0.4 | 99.1±0.3 | 87.8±7.2 | 64.5±1.4 | 96.5±1.8 | 63.4±3.4 |
| GPN - UR | 66.1±1.4 | 99.1±0.0 | 90.5±0.3 | 64.9±0.5 | **98.1**±0.6 | 70.5±2.7 |
| GPN -UR -TV | 66.3±0.8 | **99.2**±0.1 | **90.6**±0.8 | 63.6±1.2 | 98.0±0.7 | **75.6**±6.1 |

Bold numbers indicate the best model across all the uncertainty metrics for each dataset. Underlined numbers are the best results within the same model type for each dataset. n.a. means either model or metric not applicable. 'UH-0' takes class 0 as the OOD class and 'UP-1' takes class 1 as the OOD class.

Table 16: OOD Detection Result for UH (cont.)

| dataset | Houston - 2 | | | Houston -10 | | |
|---|---|---|---|---|---|---|
| | ID OA | OOD ROC | OOD PR | ID OA | OOD ROC | OOD PR |
| softmax-GCN | 65.9±1.7 | 62.0±4.9 | 4.9±0.5 | **73.4**±0.4 | 73.2±1.4 | 14.7±0.6 |
| GKDE | 68.5±n.a. | 92.3±n.a. | **56.9**±n.a. | 71.4±n.a. | 74.7±n.a. | **32.0**±n.a. |
| RGAE | n.a.±n.a. | 63.8±n.a. | 5.1±n.a. | n.a.±n.a. | 24.6±n.a. | 5.9±n.a. |
| TLRSR | n.a.±n.a. | 49.4±n.a. | 3.4±n.a. | n.a.±n.a. | 48.2±n.a. | 5.7±n.a. |
| EGCN | 70.0±0.8 | 90.6±0.1 | 16.7±0.1 | 73.0±0.3 | 76.0±0.5 | 17.0±0.3 |
| EGCN - UR | 69.7±1.6 | 91.0±2.8 | 25.4±7.7 | 73.1±0.1 | **78.3**±0.2 | 18.8±0.2 |
| EGCN - UR -TV | **70.4**±0.3 | **97.3**±0.4 | 44.9±4.2 | 73.1±0.6 | 77.1±0.4 | 18.1±0.3 |
| GPN | 64.1±0.3 | 70.1±0.6 | 6.1±0.1 | 70.9±0.7 | 58.7±1.9 | 10.5±0.4 |
| GPN - UR | 64.1±0.2 | 70.8±0.4 | 6.3±0.1 | 68.0±1.1 | 65.4±1.7 | 12.0±0.6 |
| GPN -UR -TV | 54.8±0.3 | 85.6±0.8 | 11.8±0.7 | 70.8±1.0 | 67.3±2.3 | 12.8±0.8 |

Bold numbers indicate the best model across all the uncertainty metrics for each dataset. Underlined numbers are the best results within the same model type for each dataset. n.a. means either model or metric not applicable. 'UH-2' takes class 2 as the OOD class and 'UP-10' takes class 10 as the OOD class.

Table 17: OOD Detection Result for KSC

| dataset | KSC - 5 | | | KSC - 6 | | |
|---|---|---|---|---|---|---|
| | ID OA | OOD ROC | OOD PR | ID OA | OOD ROC | OOD PR |
| softmax-GCN | 89.9±0.2 | 71.6±1.6 | 7.0±0.4 | 87.7±0.2 | 45.5±1.7 | 2.3±0.0 |
| GKDE | 87.5±n.a. | 47.0±n.a. | 5.1±n.a. | 85.0±n.a. | 43.2±n.a. | 2.2±n.a. |
| RGAE | n.a.±n.a. | 68.6±n.a. | 6.9±n.a. | n.a.±n.a. | 71.1±n.a. | 4.1±n.a. |
| TLRSR | n.a.±n.a. | 67.7±n.a. | 6.0±n.a. | n.a.±n.a. | 75.6±n.a. | 3.1±n.a. |
| TRDFTVAD | n.a.±n.a. | 64.2± 4.5 | 5.5± 0.7 | n.a.±n.a. | 67.1±8.5 | 2.5±0.7 |
| EGCN | 89.7±0.0 | 64.6±0.0 | 7.1±0.0 | 87.6±0.1 | 17.3±0.8 | 1.9±0.0 |
| EGCN - UR | 89.6±0.0 | 64.4±0.1 | 7.0±0.0 | 87.6±0.0 | 19.5±0.9 | 1.9±0.0 |
| EGCN - UR -TV | **90.0**±0.1 | **76.7**±0.5 | **12.0**±0.3 | **87.7**±0.1 | 41.9±2.8 | 2.2±0.1 |
| GPN | 86.8±1.6 | 54.3±7.4 | 4.7±0.9 | 82.6±1.9 | 26.6±16.3 | 2.0±0.3 |
| GPN - UR | 86.3±2.2 | 58.7±2.6 | 5.0±0.3 | 80.7±0.9 | 86.7±1.9 | 7.8±1.4 |
| GPN -UR -TV | 88.5±0.6 | 58.3±9.0 | 5.2±0.9 | 82.1±0.6 | **90.4**±2.5 | **20.0**±9.0 |

Bold numbers indicate the best model across all the uncertainty metrics for each dataset. Underlined numbers are the best results within the same model type for each dataset. n.a. means either model or metric not applicable. 'KSC-5' takes class 5 as the OOD class and 'KSC-6' takes class 6 as the OOD class.

Table 18: OOD Detection Result for KSC (cont.)

| dataset | KSC - 7 | | | KSC - 12 | | |
|---|---|---|---|---|---|---|
| | ID OA | OOD ROC | OOD PR | ID OA | OOD ROC | OOD PR |
| softmax-GCN | **88.0**±0.2 | 38.9±0.8 | 6.4±0.1 | 83.9±0.3 | 98.9±0.2 | 91.7±2.2 |
| GKDE | 85.8±n.a. | 88.9±n.a. | **58.7**±n.a. | 80.9±n.a. | 100.0±n.a. | 59.1±n.a. |
| RGAE | n.a.±n.a. | 18.6±n.a. | 4.8±n.a. | n.a.±n.a. | 93.0±n.a. | 56.6±n.a. |
| TLRSR | n.a.±n.a. | 67.9±n.a. | 14.7±n.a. | n.a.±n.a. | 49.9±n.a. | 9.1±n.a. |
| TRDFTVAD | n.a.±n.a. | 70.7± 2.7 | 22.6± 2.1 | n.a.±n.a. | 9.5±1.1 | 9.8±0.1 |
| EGCN | 87.8±0.3 | 93.5±0.3 | 51.1±2.2 | **84.3**±0.1 | **100.0**±0.0 | **99.9**±0.0 |
| EGCN - UR | 87.7±0.2 | **93.8**±1.0 | 53.6±4.3 | **84.3**±0.1 | **100.0**±0.0 | **99.9**±0.0 |
| EGCN - UR -TV | 87.9±0.1 | 93.7±0.8 | 56.2±6.2 | 84.2±0.1 | **100.0**±0.0 | **99.9**±0.0 |
| GPN | 86.1±1.8 | 59.0±7.8 | 9.7±1.8 | 75.8±2.6 | **100.0**±0.0 | **99.9**±0.0 |
| GPN - UR | 82.1±1.1 | 78.6±1.4 | 18.9±2.7 | 77.4±0.9 | **100.0**±0.0 | **99.9**±0.0 |
| GPN -UR -TV | 81.6±1.6 | 79.5±1.6 | 20.6±2.0 | 81.0±0.7 | **100.0**±0.0 | **99.9**±0.0 |

Bold numbers indicate the best model across all the uncertainty metrics for each dataset. Underlined numbers are the best results within the same model type for each dataset. n.a. means either model or metric not applicable. 'KSC-7' takes class 7 as the OOD class and 'KSC-12' takes class 12 as the OOD class.

Table 19: Ablation Study on EGCN2

| | UP- 6 | | | UP-7 | | |
|---|---|---|---|---|---|---|
| | ID OA | ROC | PR | ID OA | ROC | PR |
| EGCN2 | **70.7**±2.8 | 73.9±1.2 | 5.0±0.2 | 69.1±1.8 | 83.8±0.2 | 20.1±0.2 |
| EGCN2 - UR | 65.3±4.2 | **84.6**±0.3 | **7.4**±0.1 | 63.7±2.7 | **85.5**±0.1 | **22.0**±0.2 |
| EGCN2 - TV | 70.2±1.7 | 75.6±0.6 | 5.3±0.1 | **72.0**±1.4 | 84.0±0.1 | 20.4±0.2 |
| EGCN2 - UR -TV | 65.6±4.6 | 84.5±0.3 | **7.4**±0.1 | 65.6±3.7 | 85.4±0.2 | 21.9±0.3 |

## F    RELATED WORK

**Hyperspectral Imaging Analysis.**    Due to the wealth of detailed spectral information available in each pixel, hyperspectral imaging (HSI) has found widespread application in various real-world scenarios. *HSI classification* (HSIC) aims to assign a distinct class label to each pixel. In their work, (Chen et al., 2014) utilized stacked auto-encoder networks for HS image classification by leveraging dimensionally-reduced HS images obtained through principal component analysis (PCA). Another study  (Liu et al., 2017) introduced convolutional neural networks (CNNs) to effectively extract spatial-spectral features from HS images, resulting in improved classification performance. In separate work, a cascaded RNN was proposed  (Hang et al., 2019) to utilize spectral information comprehensively for achieving high-accuracy HS image classification. Furthermore,  (Hong et al., 2020) developed fusion modules that seamlessly integrate CNNs and miniGCNs in an end-to-end manner. We refer  (Ahmad et al., 2021) for a complete review of HSI classification task. *HSI spectral unmixing* decompose the image into a collection of reference spectral signatures with associated proportions, which is a non-negative matrix factorization problem  (Lee & Seung, 1999), which can be formed to blind and nonblind problem with linear mixing model (Qin et al., 2020) or extended linear mixing model  (Drumetz et al., 2019). We refer to Bhatt & Joshi (2020) for a systemic introduction. *Anomaly detection on HSI* involves detecting pixels in an image whose spectral characteristics deviate significantly from the surrounding or overall background pixels and attracts a lot of interest Deep learning-based methods can be divided into CNN-based  (Li et al., 2017), autoencoder-based  (Bati et al., 2015), GAN based  (Jiang et al., 2020) and RNN based  (Lyu & Lu, 2016). Some other techniques include manifold learning  (Lu et al., 2019) and low-rank representation  (Xie et al., 2021). A comprehensive review is conducted by Hu et al. (2022).

**Uncertainty quantification models on i.i.d inputs.**    Numerous studies have focused on developing uncertainty quantification models for data that is independent of inputs, such as images. These efforts encompass various approaches, including multi-forward pass models such as ensembles (Lakshminarayanan et al., 2017), dropout-based models  (Gal & Ghahramani, 2016), and deterministic models like Bayesian-based methods (Charpentier et al., 2022).

**Uncertainty quantification on graph data.**    As pointed out in the survey (Abdar et al., 2021), uncertainty quantification on GNN and semi-supervised learning is under-explored. Most existing models for uncertainty quantification on graphs are either dropout-based or BNN-based methods that typically drop or assign probabilities to edges. Two approaches quantified uncertainty using deterministic single-pass GNNs. One is called graph-based kernel Dirichlet distribution estimation (GKDE) (Zhao et al., 2020), which consists of evidential GCN, graph-based kernel, teacher network, dropout, and loss regularization. Another method is the GPN  (Stadler et al., 2021) model that combines PN  (Charpentier et al., 2020) and personalized page rank (PPR) message passing to disentangle uncertainty with and without network effects. In addition, a recent method (Wu et al., 2023) used standard classification loss for OOD detection on graphs together with an energy function that is directly extracted from GNN. This method is limited to OOD detection, not generally on the topic of uncertainty quantification. However, these models have feature collapsing issues and the physical mixing properties of HSI are not yet built into the models.

**Uncertainty quantification on hyperspectral image classification models.**    To the best of our knowledge, EGCN and GPN are the state-of-the-art models on the node-level uncertainty estimation for graph data. Additionally, there is currently no directly related work on uncertainty quantification for hyperspectral imaging classification. As pointed out in a recent survey (Su et al., 2021), anomaly detection is one of the most closely related and well-studied topics in HSI. Therefore, we use three anomaly detection models as baselines. However, it's important to note that while anomaly detection aims to detect pixels whose spectral characteristics deviate significantly from surrounding or background pixels, OOD detection focuses on identifying samples from distributions that are different from the (ID) training samples. Related work by He et al. (2022) investigated the confidence and calibration error for HSIC, which is a subset of aleatoric uncertainty, but it can not detect OOD pixels. Pal et al. (2022) examined the open-set recognition task, which does not provide any estimation of aleatoric uncertainty.

# G    LIMITATIONS

The UR term we introduce exhibits a strong correlation with the performance of unmixing. Specifically, its effectiveness hinges on the accuracy of the reference endmember matrix, which we assume is available in our problem formulation. Currently, we adopt the approach outlined in (Qin et al., 2020) for the endmember matrix, which serves as our reference. Nevertheless, a universally accepted criterion for assessing the quality of endmember matrices is lacking, and a possible inaccurate one could potentially undermine the positive impact of our UR term. Besides, certain categories pose challenges for unmixing models due to unpredictable noise and intricate mixtures. In such instances, our proposed UR term might not offer substantial assistance. However, we introduce the uncertainty quantification problem in the HSI domain, which is less explored but necessary in real applications. Additionally, we propose a promising direction in that we can improve deterministic uncertainty quantification models with domain knowledge, and the resulting insight obtained in this project can be extended to other application domains.

