# OpenReview forum: "Uncertainty-aware Graph-based Hyperspectral Image Classification"
_ICLR.cc/2024/Conference — ICLR 2024 poster_

### Official Review · Reviewer_6WzU · 2023-10-23

**Soundness:** 2 fair
**Presentation:** 2 fair
**Contribution:** 2 fair
**Rating:** 5
**Confidence:** 4

**Summary:**

This manuscript adapted two advanced uncertainty quantification models EGCN and GPN for quantifying epistemic and aleatoric uncertainties associated with the HSIC results. Specifically, two regularization terms are proposed to mitigate the limitations of UCE loss function.

**Strengths:**

1.	This manuscript applied uncertainty quantification models to the new field of HSIC, eliminating the limitations of the uncertainty cross-entropy loss function.
2.	Studies on the uncertainty quantification of HSIC results are extremely rare, and the author's research direction is very interesting and meaningful.
3.	This manuscript has provided clear questions and motivations, along with detailed formulae definitions and derivations.

**Weaknesses:**

1.	This manuscript is full of hyperparameters, and although the authors list the parameter choices for each model on each data set, the key parameters are not adequately discussed. For example, parametric sensitivity analysis of parameter β, λ1, λ2, and λ3 on each data set is necessary.
2.	The ablation experiments in this manuscript are inadequate, for example, models using TV regularization term alone, not including UR regularization term.
3.	Uncertainty has been widely studied in the previous works. This paper seems simply introduced the existed work for HSI classification. The experimental comparison is also insufficient.

**Questions:**

What are the limitations of UCE and the specific role of UR? Please describe them directly and clearly in the Abstract and contribution section.
	From Table Ⅰ, it can be found that the proposed uncertainty quantification frameworks do not reach the SOTA level on the misclassification detection tasks, especially the GPN-based variants, what is the purpose of these experiments?
	How computationally efficient is the use of UR and TV regularization terms on OOD and misclassification detection tasks.
	It is suggested that authors unify the meaning of bold and underlined numbers in each table. In addition, some bold an underlined numbers in Table 11-15 are incorrect. Please check the corresponding issues through the full manuscript.
	In the E.2-Detailed Result section, whether the proposed models are equally applicable to OOD detection for other classes in different datasets. Why are the PR values for class-10 in the UH dataset and class-7 in the KSC dataset below the baseline.
	There are some punctuation errors that should not be made in a paper. For example, "…64, 128, 256, respectively We…" and "…{10-4,10-3,10-2,10-1}(2) Number of… "in part D-Model Details. It is suggested to carefully check the textual details of the manuscript. In addition, what do parameters λ and λ' represent in the model.

---

> ### Author Response · Authors · 2023-11-23
> **Response to weaknesses**
>
> W1.
>
> Besides the general neural network parameters like learning rate and weight decay, our model (17) requires three specific hyperparameters denoted as  $\lambda_1, \lambda_2, \lambda_3$. Specifically, $\lambda_1$ is the trade-off weight for the uncertainty quantification framework itself, i.e. GKDE teacher for EGCN model or Entropy regularization for GPN. The other two hyperparameters $\lambda_2$ and $\lambda_3$ correspond to the weights for the proposed UR and TV terms, respectively. We conducted a sensitivity analysis on these two hyperparameters in Figure 6 by varying $\lambda_2$ and $\lambda_3$ and reporting the corresponding ROC performance. It was observed that the GPN model exhibits lower sensitivity to these parameters compared to the EGCN model. For instance, in the case of 'UP-8', the ROC metric shows stability to variations in $\lambda_2$ when $\lambda_3$ lies within the range of [1e-5, 1e-2]. Similarly for 'Up-7', the ROC metric remains consistent with changes in $\lambda_2$  when $\lambda_3$ is between [0.1, 1]. When the TV term's weight is set small, in the order of 1e-5, the model becomes highly sensitive to changes in the UR term's weight. In the EGCN framework, the model’s performance is insensitive to $\lambda_3$ when $\lambda_2$ ranges from [1e-5, 1e-4] for  `Up-8', but the performance decreases significantly when $\lambda_2$ exceeds 1e-4.
>
> W2.
>
> Thanks for the suggestion. We added this ablation design in the ablation study (Table 12). In particular, we investigate the effectiveness of all three regularization terms: R, UR, TV respectively in Equation 17. Table 12 shows that UR and TV improve the OOD detection performance individually and collaboratively. Detailed analysis can be found in Section E.1
>
> W3.
>
> Please refer to our response to Reviewer NxUY Q2. To summarize, we analyze theoretically one of the latest node-level uncertainty quantification models on graphs, i.e., EGCN, revealing the limitation that the current loss function is not powerful enough to produce good epistemic uncertainty. In detail, the current loss can not guarantee to map of OOD nodes to a region with a high epistemic uncertainty prediction on the last MLP layer on  EGCN framework. Furthermore, we propose two regularization terms based on domain-specific prior knowledge, (1) spectral features of one pixel in the hyperspectral image can be considered as a linear composition of constituent materials and the composition weight has consistent properties with the beliefs in the uncertainty quantification domain. (2) Spatial neighboring pixels tend to have similar epistemic uncertainties. Our proposed terms not only enhanced the capability of epistemic uncertainty prediction of the two latest frameworks (EGCN and GPN) on hyperspectral image classification models but also shed light on the general uncertainty-aware node-level classification models.

---

> > ### Author Response · Authors · 2023-11-23
> > **Respond to questions**
> >
> > Q1
> >
> > We explicitly address your question in abstract, introduction, and conclusion. In summary, the UCE loss is inherently designed to work with ID nodes. In the deterministic EGCN framework, a final Multilayer Perceptron (MLP) layer is responsible for reducing features to the output dimension. Only those regions that are proximate to the decision boundary determined by this last layer are identified as OOD nodes (detailed OOD detectable regions can be found in Figure 1 and Figure 2). Consequently, even if OOD pixels are distinctly separated from ID pixels in the latent feature space, the UCE loss does not promote mapping them to those OOD-detectable regions.
> >
> > Motivated by the limitation, we aim to design a mechanism to guide the neural network for better detecting the OOD nodes.  The role of the proposed UR term is to serve as a regularization that refines the solution space with the help of endmember spectra and the evidence learned by EGCN. Intuitively, it can implicitly identify OOD pixels by the material composition through unmixing reconstruction loss. This process inherently assigns high epistemic uncertainty to OOD and low epistemic uncertainty to ID.
> >
> > Q2
> >
> > Our proposed models estimate both aleatoric and epistemic uncertainty within a single forward run and measures of aleatoric uncertainty are well suited for misclassification detection while epistemic measures of uncertainty are found to be better suited
> > for detecting OOD samples [1]. Therefore, we use two tasks, i.e., misclassification detection and OOD detection, to demonstrate the effectiveness of our proposed model.
> >
> > We would like to point out that our proposed UR term is designed to enhance the prediction of epistemic uncertainty. By incorporating this UR term, the regularized models are expected to maintain their performance in detecting misclassified in-distribution data while showing improvement in OOD detection compared to the baseline framework.
> >
> > The improvements facilitated by the UR term are substantiated by the results presented in Table 1, where models augmented with the UR term, both alone and in conjunction with the TV term, outperform the original EGCN and GPN models. In addition,  across a total of 6 metrics, our uncertainty quantification framework shows better results on 4 metrics than softmax-GCN. Specifically, our model achieves better PR value and lower ROC value on UP and UH. A lower AUROC but a higher AUPR for our method indicates that our method produces more true positives among the top-ranked nodes than softmax-GCN, while softmax-GCN can separate true positives and negatives better than our method among the lower-ranked nodes. Please refer to our response to SyHe Q2 for detailed explanations about Table 1.
> >
> > Please refer to our response to SyHe W2 for more detailed explanations.
> >
> > Q3
> >
> > Compared to the parameters in the learning process, the computational complexities of UR and TV are negligible. Detailed running time and number of parameters are shown in Table 11. Thanks for the suggestion of table format, which has been corrected.
> >
> > Q4
> >
> > We follow the standard left-out class settings in the literature such as [1,2], where we randomly pick some classes as the OOD class and the remaining classes as ID. In this paper, we only consider a single OOD class and we will consider multiple OOD classes in the future. Given the potential for considerable variation in data structures across classes, we select four such configurations at random and report the weighted average and standard deviation.
> >
> > Our proposed model can be easily applied to any OOD setting. We can not guarantee that we will get the same conclusion for all the settings, but we believe that our chosen scenarios are indicative and should offer a broad insight into the model’s capabilities.
> >
> > Q5
> >
> > Please refer to the response to pixa W4.
> >
> > Q6
> >
> > Thanks for pointing out that, we have corrected the punctuation errors and symbol inconsistency.
> >
> > Reference
> >
> > [1] Zhao, Xujiang, et al. "Uncertainty aware semi-supervised learning on graph data." Advances in Neural Information Processing Systems 33 (2020): 12827-12836.
> >
> > [2]  Stadler, Maximilian, et al. "Graph posterior network: Bayesian predictive uncertainty for node classification." Advances in Neural Information Processing Systems 34 (2021): 18033-18048.

---

### Official Review · Reviewer_NxUY · 2023-10-31

**Soundness:** 2 fair
**Presentation:** 3 good
**Contribution:** 2 fair
**Rating:** 6
**Confidence:** 4

**Summary:**

The paper proposes two graph-based uncertainty quantification methods for hyperspectral image classification, incorporating advanced uncertainty quantification models (EGCN and GPN) and specific regularizations (unmixing and TV). The experimental results on three HS datasets demonstrate the advantages of the models. While the introduction of uncertainty quantification into HSIC is a valuable contribution, the overall novelty of the paper may be considered limited, and the experimental analysis could benefit from further depth and solidity.

**Strengths:**

1. The paper serves as a pioneering work in the exploration of uncertainty estimation within graph-based HSIC models.
2. The paper effectively presents a clear and well-defined motivation, along with a comprehensive theoretical analysis highlighting the limitations of prevalent uncertainty cross-entropy methods.
3. The inclusion of informative unmixing-based and TV-based regularizations in the context of HSIC is noteworthy, and the results successfully confirm the effectiveness of these proposed designs.

**Weaknesses:**

1. Recently, there has been extensive exploration of Graph-based HSIC, demonstrating its effectiveness. However, a key challenge lies in the high computation and space complexity arising from the pixel-level graph. How do the authors address this issue?
2. This paper's primary contribution lies in its incremental approach, integrating various existing works, including GNN, uncertainty, unmixing, and TV. Furthermore, the growing attention towards OOD and open-set recognition in HSI underscores the need for a compelling justification of the primary contribution.
4. The experimental results, while promising, require additional substantiation to validate the efficacy of the proposed methods. A more comprehensive set of experiments is necessary. Furthermore, the authors should delve into a more detailed analysis of the experimental outcomes.
5. Further elucidation is needed regarding the unmixing-based regularization. For instance, a detailed explanation of how the authors concurrently optimize abundance and endmembers within a graph net would enhance understanding.

**Questions:**

Please refer to the weaknesses.

---

> ### Author Response · Authors · 2023-11-23
> **Respond to W1&2**
>
> W1
>
> We will consider building the graph using super-pixels rather than every single pixel in a future direction. In this paper, we follow the same setting in [1], where we build the graph on the pixels that have known labels. Houston is the largest dataset that contains nearly 42,000 nodes, which is still manageable.
>
> We provide the complexity analysis in Table 11. Three baseline methods of anomaly detection (RGAE, TLRSR, TRDFTVAD) are implemented with Matlab and run on a desktop with Intel Core I7-9700 and 16GB memory. The remaining three (softmax-GCN, EGCN-based, GPN-based) are implemented with PyTorch and tested on a single GPU RTX4090 located on a server with AMD Ryzen Threadripper PRO 5955WX and 256GB memory.
>
> | Running Time (second) |  UP  |   UH  |  KSC  |
> |:---------------------:|:----:|:-----:|:-----:|
> |          RGAE         | 1660 | 11918 |  3899 |
> |         TLRSR         |  40  |   89  |   91  |
> |        TRDFTVAD       | 2100 |  n.a. | 10740 |
> |      softmax-GCN      |  177 |   30  |  126  |
> |       GKDE-based      |  320 |   43  |  129  |
> |       GPN-based       |  100 |   80  |   77  |
>
> | Number of parameters |   UP  |   UH  |  KSC  |
> |:--------------------:|:-----:|:-----:|:-----:|
> |         RGAE         |  n.a  |  n.a  |  n.a  |
> |         TLRSR        |  n.a  |  n.a  |  n.a  |
> |       TRDFTVAD       |  n.a  |  n.a  |  n.a  |
> |      softmax-GCN     | 14.9k | 41.8k | 24.9k |
> |      GKDE-based      | 14.8k | 41.5k | 24.8k |
> |       GPN-based      | 16.4k | 42.9k | 26.7k |
>
> W2
>
> In the Introduction, we summarized the main contributions of this paper, including theoretical analysis, uncertainty estimation framework, and two novel regularizations. We would like to elaborate on our contributions as follows,
>
> First, to the best of our knowledge, this is a pioneer work in discussing the uncertainty estimation on the graph-based HSIC models. Uncertainty estimation can be used for misclassification detection,  out-of-distribution detection (also called open set recognition in the classification task), and active learning which is not covered in this paper.  We analyzed the limitations of the existing framework EGCN and extended it by incorporating two regularizations (UR and TV).
>
> In detail, we first provide a detailed theoretical analysis of the EGCN framework optimized by UCE loss and reveal their shortbacks. Briefly, minimizing the UCE loss does not help an EGCN to learn embeddings that are capable of mapping OOD nodes into the OOD detectable region decided by the EGCN.  Then, we propose two regularization terms (UR and TV) to partially mitigate the aforementioned drawbacks. The UR term we propose utilizes the inherent link between unmixing domain and belief theory, and our TV term is the first to apply it to evidence, enabling learning of spatial relations beyond the scope of graph neural networks. Additionally, this approach may provide insights into the general graph domain, particularly for preserving features irrelevant to ID classification by exploring the underlying physical data structure.
>
>
>
> Reference
>
> [1] Hong, Danfeng, et al. "Graph convolutional networks for hyperspectral image classification." IEEE Transactions on Geoscience and Remote Sensing 59.7 (2020): 5966-5978.

---

> ### Author Response · Authors · 2023-11-23
> **Respond to W3&4**
>
> W3.
>
> Thanks for the suggestion. We added the following contents.
>
> For the experiments,
> 1. We added more ablation study results in Table 12 of Section E.1, which demonstrates the effectiveness of the proposed two regularization terms used separately and collaboratively.
> 2. We added one baseline algorithm of anomaly detection, labeled TRDFTVAD (published in a top journal of remote sensing in 2023), for UP and KSC datasets; see Tables 2,13,14,17,18.
>
> For the explanations,
> 1. For the analysis of misclassification results, please kindly refer to our response to SyHe W2.
> 2. For the analysis of the OOD detection result, we add Section E.4 to discuss the rationale and influence of the proposed regularizations (UR and TV) on the performance of uncertainty quantification.
>
> For theoretical analysis,
> 1. We add Theorem 3 in Appendix A.3 and reveals a limitation of training the ENN model (a generalized version of EGCN for independent inputs) solely with the UCE loss function: it might predict constant overall evidence for ID data points while erroneously assigning high overall evidence (or low vacuity) to OOD data points. By incorporating the UR regularization term into the UCE loss function and under certain additional assumptions, our analysis shows that the modified ENN model predicts lower overall evidence for OOD data points than for ID data points.
>
> W4.
>
> In Section 3.2, we assume the endmembers for ID material are known and the spectra signature of OOD material is unknown. Practically, we obtain ID endmembers a priori using a blind hyperspectral unmixing model [1]. During model optimization, we iteratively fine-tune the abundance coefficient and the endmember for OOD material. Initially, all parameters are randomly initialized. We first optimize the neural network parameters, keeping the OOD endmember value fixed, using our custom regularized learning function (Equation 17). Once the abundance coefficients for both ID and OOD materials are predicted by the neural network, we apply the unmixing loss (Equation 10) to determine the optimal OOD endmember, as analytically described in Equation 12. This process is repeated until convergence is achieved. We added a pseudo algorithm in the last paragraph of Appendix D.
>
> Reference
>
> [1] J. Qin et al., "Blind Hyperspectral Unmixing Based on Graph Total Variation Regularization," in IEEE Transactions on Geoscience and Remote Sensing, vol. 59, no. 4, pp. 3338-3351, April 2021, doi: 10.1109/TGRS.2020.3020810.

---

### Official Review · Reviewer_WkZQ · 2023-11-06

**Soundness:** 3 good
**Presentation:** 2 fair
**Contribution:** 3 good
**Rating:** 6
**Confidence:** 2

**Summary:**

This paper introduces a novel graph-based framework aimed at quantifying uncertainty in the field of HSIC. The authors analyze the limitations of the ENN models based on the UCE loss. To alleviate these limitations, this paper leverages the inherent physical properties of HS data and applies edge-preserving regularization techniques to facilitate the propagation of evidential information within the spatial domain, as a result, two regularization terms UR and evidence-based TV are proposed. The experiments on three datasets to demonstrate the effectiveness of the proposed regularizations.

**Strengths:**

1.	This paper introduces a graph-based uncertainty quantification framework for HSIC and presents UR and TV regularization methods based on the characteristics of HS data.
2.	The work on graph-based uncertainty quantification in the field of HSIC is novel and appears to be effective based on the provided experimental results.
3.	The research is well-motivated, theoretically grounded, and explores a graph-based uncertainty quantification framework for HSIC models, aligning with the characteristics of hyperspectral imaging.

**Weaknesses:**

1.	Some minor details in the writing require attention and revision. For instance, in the last paragraph of the Introduction, it should use the abbreviation 'TV' for 'total variation.' In the Conclusion, there is a shift from present tense to past tense, and it's essential to maintain consistency.
2.	In Table 1, the proposed methods show Misclassification ROC scores lower than those of softmax-GCN on the UP and UH datasets, indicating that the best results have not been achieved. The paper lacks relevant explanations, and a more comprehensive and in-depth experimental analysis is needed.
3.	In the comparative experiments, it's worth noting that there are instances where the introduction of TV leads to a decrease in results. For example, in Table 11 for OOD PR metric of UP-4, GPN-UR scores 99.1 and GPN-UR-TV scores 98.9. This raises questions about the robustness of these methods. It would be beneficial to provide relevant explanations and analyses. Additionally, it might be worthwhile to include separate experimental results for EGCN-TV and GPN-TV.
4.	The compared baselines need to be enriched. There are some more recent methods published that should be introduced and compared, such as " Hyperspectral Anomaly Detection Based on Tensor Ring Decomposition With Factors TV Regularization ", " Hyperspectral anomaly detection based on variational background inference and generative adversarial network".

**Questions:**

1.	What does "ID" mean in this paper?
2.	Is 'Mis' an abbreviation for Misclassification in Table 1? It should be explained in the paper.
3.	Do the numbers following the dataset names in Table 11 to Table 16, such as '4' in 'UP-4,' represent the class numbers? It should be clarified in the paper for better understanding.

---

> ### Author Response · Authors · 2023-11-23
>
> W1.
>
> Thanks for pointing this out. We have made these changes in the latest version of the manuscript.
>
> W2.
>
> Please refer to our detailed response to Reviewer SyHe W2. In summary, our proposed UR and TV term aims to enhance the capability of epistemic uncertainty prediction while misclassification detection using aleatoric uncertainty as the measure. Therefore, the regularized models are expected to maintain their performance in detecting misclassified ID data while showing improvement in OOD detection(measured by the epistemic uncertainty) compared to the baseline framework. Table 1 shows that including the UR term alone and in conjunction with the TV term outperforms the original EGCN and GPN models in most cases.
>
> For this mentioned example, our model achieves better PR value and lower ROC value on UP and UH compared to softmax-GCN. A lower AUROC but a higher AUPR for our method indicates that our method produces more true positives among the top-ranked nodes than softmax-GCN, while softmax-GCN can separate true positives and negatives better than our method among the lower-ranked nodes. Besides, Table 1 shows that for the majority of settings (4/6 metrics), our uncertainty quantification framework has better performance than the softmax-GCN, and incorporating our proposed UR/TV term can get better or at least comparable results for most settings.
>
> W3.
>
> Thanks for pointing it out. Please refer to our response to Reviewer pixa W4 for explanations of OOD detection results and we added case studies related to the TV and UR term in Section E.4. We added Section E.1 and Table 12 as a throughout ablation study for EGCN-TV and GPN-TV.
>
> For this specific case on 'UP-4’, the GPN-UR-TV model did not show obvious improvement on GPN-UR. This may be because the TV term is designed to smooth the epistemic uncertainty across spatial neighbors, and it may bring more gain when the pixels belonging to the ID or OOD are more concentrated, i.e. neighboring nodes tend to have similar epistemic uncertainty. Under the opposite scenario, the TV term may maintain the model performance and it may show some slightly better or worse results due to the model randomness.  Additionally, since the baseline GPN model already achieves near-perfect OOD detection on 'UP-4', improving upon this high performance may be more challenging compared to models with poorer OOD detection results.
>
> W4.
>
> Thanks for suggesting the reference. We added the mentioned baseline called 'TRDFTVAD' for the first paper " Hyperspectral Anomaly Detection Based on Tensor Ring Decomposition With Factors TV Regularization. " We did the same hyperparameter search as the original paper with 5 random runs on two datasets 'UP' and 'KSC'. We skipped the Houston dataset due to the running time and memory required by the large spatial dimension. For the second paper, we cannot get the source code from the authors in the past 12 days, thus unable to add the comparison results.
> The OOD detection results from 'TRDFTVAD' are listed as follows and have been added in Table 2, 13,14,17,18. All variations of our proposed models perform better than TRDFTVAD in most scenarios.
>
> ### OOD detection on UP dataset
> | OOD setting | OOD-ROC | OOD-ROC std | OOD-PR | OOD-PR std |
> |:-----------:|:-------:|:-----------:|:------:|:----------:|
> |     UP-4    |  0.9833 |    0.0011   | 0.6134 |   0.0237   |
> |     UP-6    |  0.5290 |    0.0243   | 0.0234 |   0.0012   |
> |     UP-7    |  0.5787 |    0.0139   | 0.0905 |   0.0027   |
> |     UP-8    |  0.9254 |    0.0105   | 0.1234 |   0.0127   |
>
> ### OOD detection on KSC dataset
> | OOD setting | OOD-ROC | OOD-ROC std | OOD-PR | OOD-PR std |
> |:-----------:|:-------:|:-----------:|:------:|:----------:|
> |    KSC-5    |  0.6422 |    0.0449   | 0.0547 |   0.0070   |
> |    KSC-6    |  0.6706 |    0.0848   | 0.0251 |   0.0068   |
> |    KSC-7    |  0.7072 |    0.0286   | 0.2261 |   0.0209   |
> |    KSC-12   |  0.9254 |    0.0105   | 0.1234 |   0.0127   |
>
> Q1.
>
> It means in-distribution, which is defined on Page 6. We will define it when it is first used in the Introduction.
>
> Q2.
>
> Yes. For clarity, we removed “mis” in front of ROC/PR in Table 1.
>
> Q3.
>
> We explained in the caption of Table 11 that  ‘UP-4’ takes class 4 as the OOD class and ‘UP-6’ takes class 6 as the OOD class.

---

### Official Review · Reviewer_pixa · 2023-11-07

**Soundness:** 2 fair
**Presentation:** 2 fair
**Contribution:** 3 good
**Rating:** 5
**Confidence:** 2

**Summary:**

This paper introduces the uncertainty quantification to the Hyperspectral imaging classification (HSIC), i.e. epistemic and aleatoric uncertainties. Specifically, the paper theoretically analyzes the limitation of uncertainty cross-entropy (UCE) loss in evidential graph convolution neural networks (EGCN) and proposes two regularization terms to deal with this limitation. Experiments are performed on three real-world HSIC datasets.

**Strengths:**

1. This paper analyzes the reason why the UCE-based MLP layer in EGCN cannot accurately obtain evidence predictions.
2.  A solution for the above problem is also provided.

**Weaknesses:**

1. The rationality of the Unminxing-based Regularization part is insufficient. First, the HSIC problem containing noise will greatly affect the optimization results of Eq. (10). Secondly, even considering a simple case without noise, whether the ID class and OOD class can be split as Eq. (10) and its rationality remain to be verified.
2. The unmixing-baed regularization (UR) term, i.e. Eq. (11), is designed heuristically, and its rationality has not been fully explained. Specifically, the optimization trends of $\mathbf{b}^{i}(\mathbf{\theta})$ and $u^{i}(\mathbf{\theta})$ explained in the main text are consistent with the hyperspectral unmixing problem, but I think the rational explanation of this part is insufficient.
3. The proposed evidence-based total variation regularization term is incremental. And Proposition 2 is a relatively simple and intuitive conclusion based on Eq. (13).
4. The proposed improvements have not been consistently verified in experimental performance.
5. The compared softax-GCN method is an earlier method, and the author can try some of the latest methods.

**Questions:**

Could the author expand their analysis of the limitations and solutions related to the UCE model and apply it to the design of the EGCN model? This would provide a more coherent and insightful discussion.

---

> ### Author Response · Authors · 2023-11-23
> **Response to weakness 1**
>
> W1
>
> It is true that there are many nuisance factors that affect the optimization results by minimizing (10), and yet it is still widely used in hyperspectral unmixing if we replace the beliefs/vacuity with the abundance coefficients. Since this paper focuses on uncertainty quantification, it is in fact our contribution to approximate the abundance coefficients for ID and OOD nodes by beliefs and vacuities, respectively. In what follows, we explain our motivation for such approximation and its rationale from three perspectives.
>
> First, belief and vacuity **share similar properties** to abundance coefficients, as described in the paragraph after Eq. (10). Specifically, OOD nodes are generally characterized by elevated vacuity uncertainty and possess a substantially large abundance coefficient pertinent to the corresponding material that is not in the training set. In parallel, ID nodes can be distinguished by large class-specific beliefs and bear class-specific abundance coefficients that align with the veritable ID class. Moreover, both belief/vacuity and abundance coefficient conform to a uniform scale owing to their sum-to-one property.
>
> Second, there is a **parallelism in the physical interpretations** of abundance coefficients to the concepts of belief/vacuity. The abundance coefficient represents the proportion of a specific material, with the spectral feature of a single pixel being a composite (weighted by the proportions of all materials present). In a similar vein, the concept of belief quantifies the extent of evidential support derived from the data, indicating the likelihood of a sample's classification into a particular class. The belief essentially measures the accumulation of observations that corroborate the class assignment of a sample.
>
> Third, there is an **underlying assumption** for the UR term to improve the OOD detection. Specifically, Equation (10) assumes that the feature vector corresponding to an ID node is a linear combination of ID endmembers; and consequently, if a feature vector $\textbf{x}$ is orthonormal to the space that is spanned by $M$, then it is indicative of an OOD node. (10) holds particularly true in scenarios where the OOD spectral signature is orthogonal to $M$. Conversely, if the OOD spectral is nearly parallel to one of the ID endmembers, then the UR term does not help to distinguish the OOD node from this ID material.
>
> We intend to **empirically verify this assumption** by using cosine similarity to measure how close two endmembers’ spectra are in the sense that 0 means they are orthogonal (dissimilar) and 1 means they are parallel (identical).  We take the 'UP-6' and 'KSC-5' as two illustrative examples. From Table 15 and Table 17, we note that the UR term yields an improvement of 18\% and 40.4\% in ROC for 'UP-6', whereas the improvement for  'KSC-5' is relatively modest, at -0.2\% and 4.4\% for the EGCN and GPN frameworks respectively. The difference in UR's improvement lies in the difficulty of identifying the OOD class. We present the distribution of cosine distances between OOD nodes and ID nodes of UP-6 and KSC-5 in Figure 7.  For `UP-6', class 'Bituman' (class 6) displays a relatively low degree of similarity (approximately 0.8) compared to nearly 0.9 in KSC-5.  This verifies that the UR-based regularization modeled in Equation (10) can improve OOD detection performance under the assumption that OOD spectral should be as orthogonal to the ID feature space as possible.
>
> We also performed **a theoretical analysis, as elaborated in Appendix A.3**, to validate the effectiveness of the UR regularization term in enhancing the epistemic quantification capabilities of a simple 1-MLP ENN model for OOD detection. In our analysis, we assume that ID and OOD data points are derived from a linear mixture model, with the OOD signature vector being orthogonal to the ID materials' signature vectors. Our findings reveal that using only the UCE loss function for training the ENN model leads to a limitation: the model may predict a constant overall evidence for an ID data point, but it could assign inappropriately high overall evidence (or equivalently, very low vacuity) to an OOD data point. This is problematic since the overall evidence for OOD data should ideally be low, leading to the model's ineffectiveness in OOD detection. However, by incorporating the UR regularization term into the UCE loss function and under certain additional assumptions, our analytical results show that the enhanced ENN model predicts lower overall evidence for OOD data points compared to ID data points. Consequently, this modified ENN model exhibits improved effectiveness in detecting OOD data points over the version trained solely with the UCE loss.

---

> ### Author Response · Authors · 2023-11-23
> **Response to weakness 2&3&5**
>
> W2
>
> Eq (11) is obtained by Eq (10), after which we explained the three-folded rationale of such approximations, ${\bf b}^i(\bf{\theta})\approx{\bf v}^i(\bf{\theta})$ and $u^i(\bf{\theta}) \approx v_o^i(\bf{\theta}) $. Here we elaborate on some details. For ID nodes, its vacuity $u^i=0$ and the class probability can be regarded as the abundance coefficient; such an assumption has been explored in hyperspectral unmixing by [1]. It further follows from [2] that the beliefs are analogous to class probability, which justifies that ${\bf b}^i(\bf{\theta})\approx{\bf v}^i(\bf{\theta})$. On the other hand, the vacuity for an OOD node is close to one and its belief is close to zero, implying that the abundance coefficient $v_o^i(\bf{\theta})$ should be close to one and ${\bf v}^i(\bf{\theta})$ be close to ${\bf 0}$.
>
> W3
>
> To the best of our knowledge, this is the first time TV is applied to evidence for hyperspectral unmixing, which is obtained through a deep learning architecture.
> It is true that the conclusion of Prop 2 is simple, but it is worthwhile to explicitly point out these two desired properties of the UR term to provide additional rationale for incorporating the UR term in the learning process.
>
> W5
>
> To the best of our knowledge, EGCN and GPN are two state-of-the-art models for node-level uncertainty quantification on graphs and there is no work on uncertainty quantification of hyperspectral image classification tasks. For the baselines, we also consider the three latest anomaly detection models, considering that it is the most related task that is well-studied in the hyperspectral image classification task. The main reason we compared our models with softmax-GCN is because EGCN uses the GCN as the feature extraction.  GPN is considered the MLP layer for feature learning and GNN (APPNP in the model design) is for label propagation in the last step.
>
> To investigate the impact of different encoder backbones on OOD detection, we apply a more recent and widely cited deep GNN architecture [3]  (with over 1K citations) as the backbone for the EGCN framework, termed EGCN2. The results, as shown in the following Table (Table 19 in the manuscript), align with those obtained using GCN as the encoder, where the UR term notably enhances the ROC and PR metrics for OOD detection. While the TV term yields comparable OOD detection, it demonstrates improved in-distribution (ID) classification performance. Furthermore, a comparison with Table 12 reveals that GCN2 exhibits superior OOD detection on the 'UP-7' dataset but shows reduced effectiveness on 'UP-6'. With the time lime limit, we only consider the same OOD settings as our ablation study.
>
> |      UP-6      |    ID OA     |      OOD-ROC     |      OOD-PR     |
> |:--------------:|:------------:|:------------:|:-----------:|
> |      EGCN2     | 70.7$\pm$2.8 | 73.9$\pm$1.2 | 5.0$\pm$0.2 |
> |   EGCN2 - UR   | 65.3$\pm$4.2 | 84.6$\pm$0.3 | 7.4$\pm$0.1 |
> |   EGCN2 - TV   | 70.2$\pm$1.7 | 75.6$\pm$0.6 | 5.3$\pm$0.1 |
> | EGCN2 - UR -TV | 65.6$\pm$4.6 | 84.5$\pm$0.3 | 7.4$\pm$0.1 |
>
> |      UP-7      |    ID OA     |      OOD-ROC     |      OOD-PR      |
> |:--------------:|:------------:|:------------:|:------------:|
> |      EGCN2     | 69.1$\pm$1.8 | 83.8$\pm$0.2 | 20.1$\pm$0.2 |
> |   EGCN2 - UR   | 63.7$\pm$2.7 | 85.5$\pm$0.1 | 22.0$\pm$0.2 |
> |   EGCN2 - TV   | 72.0$\pm$1.4 | 84.0$\pm$0.1 | 20.4$\pm$0.2 |
> | EGCN2 - UR -TV | 65.6$\pm$3.7 | 85.4$\pm$0.2 | 21.9$\pm$0.3 |
>
>
> Reference
>
> [1] Chen, Bohan, et al. "Graph-based Active Learning for Nearly Blind Hyperspectral Unmixing." IEEE Transactions on Geoscience and Remote Sensing (2023).
>
> [2] ​​Jsang, Audun. Subjective Logic: A formalism for reasoning under uncertainty. Springer Publishing Company, Incorporated, 2018.
>
> [3] Chen, Ming, et al. "Simple and deep graph convolutional networks." International conference on machine learning. PMLR, 2020

---

> ### Author Response · Authors · 2023-11-23
> **Response to W4 (analysis of experimental result)**
>
> W4
>
> For misclassification detection, please refer to a detailed response to Syhe Q2. Briefly, our proposed UR and TV term aims to enhance the capability of epistemic uncertainty prediction and misclassification detection using aleatoric uncertainty as the measure. Therefore, the regularized models are expected to maintain their performance in detecting misclassified ID data while showing improvement in OOD detection(measured by the epistemic uncertainty) compared to the baseline framework. Table 1 shows that including the UR term alone and in conjunction with the TV term outperforms the original EGCN and GPN models in most cases, which implies the effectiveness of the proposed regularizations.
>
> For OOD detection, we follow the same setting of the left-out class OOD as in [1] [2], and we randomly pick one class as the OOD class. Considering that different choices of the OOD class may exhibit disparate performance on the OOD detection, we randomly generate 4 OOD settings for each dataset and it may be difficult for the regularization terms to always have the best results across all scenarios.  We add observations on KSC in Appendix E.5 showing that the inclusion of UR and TV terms consistently enhances OOD detection performance on the KSC dataset under the specified experimental settings. We also add an ablation study to investigate the effectiveness of all three regularization terms: R, UR, TV respectively in Equation 17 on UP in  Appendix E.1 and Table 12 shows that UR and TV improve the OOD detection performance individually and collaboratively.
>
> Then we discuss the reasons for scenarios with low performance of UR or TV terms on OOD detection briefly and we add a detailed analysis in Appendix E.4. We provide an analysis of UR term in our response to W1 and TV term here.
>
> Comparing 'UH-2' and 'UH-10' in Table 16, the TV term brings significant improvements on 'UH-2'.  This is attributed to the TV term's smoothing effect on predicted total evidence (inverse of epistemic uncertainty) across spatial neighbors.  Class 2 ('artificial turf'), being more clustered than Class 10 ('sidewalks') as depicted in Figure 8, benefits more from the TV term due to the higher likelihood of spatial neighbors sharing similar ground truth evidence.
>
> When pixels of a specific class are sparsely distributed, the addition of the TV term may not significantly enhance OOD detection, often resulting in comparable outcomes instead. This could be due to various factors contributing to the combined TV and UR terms performing slightly less effectively in OOD detection compared to using the UR term alone. (1) There often exists a trade-off between OOD detection performance and ID classification performance, as discussed in a recent work [3]. In the case of `EGCN-UR-TV' applied to the 'Houston-1' dataset, we observe a lower ROC but a higher ID Overall Accuracy (OA). (2)Examining the combination of ROC and PR provides a more comprehensive perspective, especially for imbalanced scenarios. 'GPN-UR-TV' has a lower ROC but a higher PR value for 'KSC-5' in Table 15. (3) Model randomness may lead to a reasonable performance variance.
>
> We would also like to highlight the distinctions between the two frameworks, EGCN and GPN, as they may exhibit divergent performance patterns. Despite both frameworks utilizing the same UCE loss function, EGCN employs a discriminative boundary for prediction, whereas GPN employs a generative model for density estimation. Consequently, these frameworks encounter distinct challenges during the training phase.
>
> Reference
>
> [1] Zhao, Xujiang, et al. "Uncertainty aware semi-supervised learning on graph data." Advances in Neural Information Processing Systems 33 (2020): 12827-12836.
>
> [2] Stadler, Maximilian, et al. "Graph posterior network: Bayesian predictive uncertainty for node classification." Advances in Neural Information Processing Systems 34 (2021): 18033-18048.
>
> [3] Teney, Damien, et al. "ID and OOD performance are sometimes inversely correlated on real-world datasets." arXiv preprint arXiv:2209.00613 (2022).

---

> ### Author Response · Authors · 2023-11-23
> **Respond to question**
>
> Question
>
> In Theorem 1, we illustrate that solely minimizing the UCE loss does not effectively train the last MLP layer of the EGCN model to map OOD regions (as depicted in the light-green areas in Figs. 1 and 2) into EGCN's detectable OOD region near the decision boundary. We want to highlight that it is difficult to conduct a direct analytical analysis on the effectiveness of the UCE loss in EGCN training without any simplifications on the EGCN architecture.
>
> To demonstrate that our proposed Unmixing-based Regularization (UR) term in Section 3.2 can alleviate the limitations of the UCE loss function in learning an EGCN model, we conducted a theoretical analysis based on a predefined linear mixture model, as detailed in Appendix A.3. This analysis demonstrates the efficacy of the UR term in boosting the epistemic quantification of a simplified 1-MLP ENN model for OOD detection. We make the assumption that both ID and OOD data points originate from a linear mixture model, with the OOD signature vector orthogonal to the ID materials' signature vectors. Our results indicate a critical limitation when using only the UCE loss function: the model may consistently predict the same overall evidence for an ID data point, but it might inaccurately assign extremely high overall evidence to an OOD data point. This misrepresentation, indicating a low level of uncertainty for OOD data, undermines the model's OOD detection capability. However, integrating the UR term into the UCE loss function, along with certain assumptions, leads to a more effective outcome. Our analysis reveals that the modified ENN model, enhanced with the UR term, predicts lower overall evidence for OOD data compared to ID data, thereby significantly improving its ability to detect OOD data points compared to a model trained only with the UCE loss.

---

### Official Review · Reviewer_SyHe · 2023-11-08

**Soundness:** 4 excellent
**Presentation:** 2 fair
**Contribution:** 3 good
**Rating:** 6
**Confidence:** 3

**Summary:**

This paper analyzes the limitations of the ENN model based on UCE loss. Additionally, the author introduces two regularization terms to alleviate these constraints, and the proposed regularization terms exhibit significant improvements on some datasets.

**Strengths:**

1. This article offers rich and comprehensive theoretical derivations, with strong formal proofs for the proposed regularization terms and UCE loss analysis, which makes the results quite convincing.
2. The motivation behind the article is clear, and the analysis is praiseworthy.
3. The proposed method has been tested on multiple benchmarks.
4. The entire manuscript is well-structured and logically organized.

**Weaknesses:**

1. The experimental section of this manuscript appears to be less robust compared to the design and discussion of the methodology.
2. The two proposed regularization terms, UR and TV, do not perform well in Table 1. I'm curious about the reasons behind this.
3. There is a lack of comparative analysis in the experimental section. I'm interested in understanding how the results compare to the latest methods.
4. The ablation experiments directly rely on the results from Tables 1 and 2 for analysis. Perhaps the author could benefit from incorporating more varied ablation designs.

**Questions:**

Please see the Weaknesses.

---

> ### Author Response · Authors · 2023-11-23
>
> W1.
>
> Thanks for your suggestion. We added more experiments in the revised manuscript, summarized as follows,
> 1. We added more ablation study results in Table 12 of Section E.1, which demonstrates the effectiveness of the proposed two regularization terms used separately and collaboratively.
> 2. We added one baseline algorithm of anomaly detection, labeled TRDFTVAD (published in a top journal of remote sensing in 2023), for UP and KSC datasets; see Tables 2, 13-14, and 17-18.
> 3.  We present hyperparameter sensitivity analysis in Figure 6, which implies that GPN-based models tend to be less sensitive to hyperparameters compared to EGCN.
> 4. We also add Section E.4 to discuss the rationals and influence of the proposed regularizations (UR and TV) on the performance of uncertainty quantification.
>
> W2.
>
> According to the “No Free Lunch” theory, there is no one model that works best for every situation. Table 1 shows that including the UR term alone and in conjunction with the TV term outperforms the original EGCN and GPN models in most cases, which implies the effectiveness of the proposed regularizations.
>
> In addition, among the six measures in Table 1, our uncertainty quantification framework is better than softmax-GCN in four of them. Specifically, our model achieves better PR value and lower ROC value on UP and UH. A lower AUROC but a higher AUPR for our method indicates that our method produces more true positives among the top-ranked nodes than softmax-GCN, while softmax-GCN can separate true positives and negatives better than our method among the lower-ranked nodes.
>
> Moreover, our proposed UR and TV regularizations are designed to improve the epistemic uncertainty. In particular, the UR term aims to assign high epistemic uncertainty for pixels containing OOD materials, while TV is applied to the total evidence, which is essentially the inverse of epistemic uncertainty.  By incorporating UR or TV terms, the regularized models are expected to maintain their performance in detecting misclassified ID nodes (measured by aleatoric uncertainty) while improving the capabilities of OOD detection (measured by epistemic uncertainty) compared to the baseline.
>
> Lastly, we would like to point out that the GPN paper [1] also took the “No Free lunch” setting into consideration and reported a similar behavior in their baseline comparison, please refer to Tables 10-11 in [1] regarding misclassification detection where the GPN model gives the best ROC only on 1 out of 8 datasets and the best PR on 4 out of 8 datasets.
>
> W3.
>
> To the best of our knowledge, EGCN and GPN are the state-of-the-art models on the node-level uncertainty estimation for graph data and there is no directly related work on uncertainty quantification for hyperspectral imaging classification. As pointed out in a recent survey [2], anomaly detection is one of the most related and well-studied topics in HSI. Therefore we use three anomaly detection models as baselines. However, anomaly detection is different from OOD detection, as it aims to detect pixels whose spectral characteristics deviate significantly from surrounding or background pixels, while OOD detection involves identifying samples from disparate distribution than ID samples in the training set.
>
> Other related works we can find include [3] and [4]. [3] investigated the confidence and calibration error for HSIC, which is a subset of aleatoric uncertainty, but it can not detect OOD pixels. [4] examined the open-set recognition task, which does not provide any estimation of aleatoric uncertainty. We added these references in Section F (Related Work) under the category of “Uncertainty quantification on hyperspectral image classification models.”
>
> W4.
>
> We added Table 12 for a throughout ablation study and a detailed analysis is described in Section E.1. In particular, we investigate the effectiveness of all three regularization terms: R, UR, and TV, respectively in Equation 17. Table 12 shows that UR and TV improve the OOD detection performance individually and collaboratively.
>
> Reference
>
> [1] Stadler, Maximilian, et al. "Graph posterior network: Bayesian predictive uncertainty for node classification." Advances in Neural Information Processing Systems 34 (2021): 18033-18048.
>
> [2] Su, Hongjun, et al. "Hyperspectral anomaly detection: A survey." IEEE Geoscience and Remote Sensing Magazine 10.1 (2021): 64-90.
>
> [3] He, Xin, Yushi Chen, and Lingbo Huang. "Toward a Trustworthy Classifier With Deep CNN: Uncertainty Estimation Meets Hyperspectral Image." IEEE Transactions on Geoscience and Remote Sensing 60 (2022): 1-15
>
> [4] Pal, Debabrata, et al. "Few-shot open-set recognition of hyperspectral images with outlier calibration network." Proceedings of the IEEE/CVF Winter Conference on Applications of Computer Vision. 2022.

---

> ### Author Response · Authors · 2023-11-23
> **Ablation Study Table**
>
> We list the ablation study on UP-6 and UP-7 in the following two tables.  Detailed analysis can be found in Appendix E.1.
>
> |     UP-6          |     ID OA      |       OOD ROC       |       OOD PR       |
> |---------------|:--------------:|:---------------:|:--------------:|
> |    UCE-only   | 77.16$\pm$0.62 |  72.66$\pm$1.10 |  4.08$\pm$0.15 |
> | UCE-AT (EGCN) | 74.86$\pm$2.37 |  74.82$\pm$4.45 |  4.50$\pm$0.67 |
> |    EGCN-UR    | 74.32$\pm$1.19 |  **90.79**$\pm$0.42 | 10.94$\pm$0.40 |
> |    EGCN-TV    | 77.13$\pm$2.15 |  83.96$\pm$0.74 |  6.63$\pm$0.28 |
> |   EGCN-TV-UR  | **77.37**$\pm$1.78 |  89.03$\pm$1.02 |  9.33$\pm$0.74 |
> |---------------|:--------------:|:---------------:|:--------------:|
> |      UCE-only      | 66.92$\pm$3.95 |  30.60$\pm$7.65 |  1.69$\pm$0.15 |
> |  UCE-ER (GPN) | 66.11$\pm$7.32 | 39.40$\pm$16.80 |  2.06$\pm$0.63 |
> |     GPN-UR    | 54.67$\pm$2.44 |  90.19$\pm$0.46 | 10.78$\pm$0.34 |
> |     GPN-TV    | 66.50$\pm$1.56 |  61.08$\pm$4.53 |  3.07$\pm$0.41 |
> |   GPN-TV-UR   | 54.73$\pm$1.39 |  90.39$\pm$0.34 | **11.31**$\pm$0.37 |
>
> |         UP-7      |     ID OA      |     OOD  ROC       |    OOD   PR       |
> |---------------|:--------------:|:---------------:|:--------------:|
> |    UCE-only   | 75.03$\pm$3.08 |  80.70$\pm$1.87 | 21.73$\pm$2.88 |
> | UCE-AT (EGCN) | **75.91**$\pm$2.23 |  80.51$\pm$2.13 | 21.72$\pm$3.56 |
> |    EGCN-UR    | 72.31$\pm$1.70 |  86.66$\pm$1.71 | 26.42$\pm$3.65 |
> |    EGCN-TV    | 75.50$\pm$1.24 |  79.81$\pm$2.98 | 20.88$\pm$4.06 |
> |   EGCN-TV-UR  | 75.05$\pm$3.17 |  86.78$\pm$1.48 | 26.01$\pm$3.32 |
> |---------------|:--------------:|:---------------:|:--------------:|
> |    UCE-only   | 64.36$\pm$5.82 |  71.58$\pm$8.50 | 14.16$\pm$3.94 |
> |  UCE-ER (GPN) | 64.94$\pm$3.72 | 74.58$\pm$12.10 | 16.51$\pm$5.07 |
> |     GPN-UR    | 58.73$\pm$3.71 |  90.58$\pm$1.97 | 32.54$\pm$5.22 |
> |     GPN-TV    | 61.00$\pm$3.31 |  82.33$\pm$1.58 | 19.52$\pm$0.98 |
> |   GPN-TV-UR   | 66.08$\pm$4.39 |  **92.89**$\pm$0.17 | **42.47**$\pm$0.92 |
>
> Observations:
> 1. The proposed UR term significantly improves the accuracy of OOD detection compared to other baseline models. Specifically, in the context of UP-6, the UR term augments ROC by 15.97\% and 50.79\%, and PR by 6.44\% and 8.72\% for EGCN and GPN respectively. For UP-7, improvements of 6.15\% and 16\% in ROC together with 4.7\% and 16.03\% in PR are observed for EGCN and GPN, respectively.
> 2. The TV regularization term also enhances OOD detection in overall, with improvements ranging from 1.25\% to 21.68\% in terms of ROC and 1.01\% to 4.18\% in terms of PR. There is a slight decrease within 1\% on EGCN for `UP-7' and it may be due to reasonable randomness. (3) Models employing both TV and UR terms generally deliver the best results, except for the EGCN model on UP-6. Although the EGCN-TV-UR performs slightly lower (within 1\%) ROC and PR than the EGCN-UR, the former achieves better ID OA with 3.05\% compared to the latter.

---

### Author Response · Authors · 2023-11-23

We thank all the reviewers for carefully reading our manuscript and providing constructive feedback. We have revised our paper accordingly with changes marked in blue for the ease of the reviewers. Here we would like to summarize these major changes we conducted during the rebuttal period:
1. We theoretically verified that incorporating the UR term can refine the solution space when detecting the OOD nodes; please refer to Theorem 3.
2. The empirical justifications of the proposed UR and TV regularization terms are given in Section E.4.
3. We added an additional baseline model for HSI anomaly detection, called TRDFTVAD. See its results added in Table 2, 13-14, 17-18.
4. We added a comprehensive ablation study on the two proposed regularization terms, UR and TV; please refer to Section E.1 and Table 12.
5. We added complexity analysis and runtime comparison in Table 11 as well as the sensitivity analysis on parameters $\lambda_2$ and $\lambda_3$ in Figure 6.

Since both theoretical analysis and additional experiments are time-consuming tasks, we manage to put together a rebuttal letter by the deadline.

---

### Meta-Review · Area_Chair_XZnP · 2023-12-02

**Metareview:**

The paper proposed an uncertainty-aware graph-based hyperspectral image classification. Specifically, the paper analyzes theoretically the limitations of a popular uncertainty cross-entropy (UCE) loss function when learning EGCNs for epistemic uncertainty estimation, and proposed two regularization terms to mitigate the limitations, Unmixing-based Regularization and Evidence-based Total Variation Regularization.

The AC has thoroughly read all the comments from the reviewers, the authors’ responses, and carefully read the paper. The main concerns raised by the reviewers revolve around the experimental performance compared to previous methods, the introduction of hyperparameters, inadequate ablation experiments, and some minor details in the writing that require attention and revision. However, the AC acknowledges that the theoretical analysis and the design of the unmixing-based regularization in the paper are both solid.

**Justification For Why Not Higher Score:**

I agree with the reviewer’s concerns.

**Justification For Why Not Lower Score:**

The method has highlights in both theoretical analysis and new loss design.

---

### Decision · Program_Chairs · 2024-01-16

Accept (poster)